# NOT ALL TASKS ARE BORN EQUAL: UNDERSTANDING ZERO-SHOT GENERALIZATION

**Jing Zhou**[1]**, Zongyu Lin**[1]**, Yanan Zheng**[2]**, Jian Li** [*1]**, Zhilin Yang** [*134]
[1]Institute for Interdisciplinary Information Sciences (IIIS), Tsinghua University
[2]Department of Computer Science and Technology, Tsinghua University
[3]Shanghai Artificial Intelligence Laboratory, [4]Shanghai Qi Zhi Institute
zhouj18@mails.tsinghua.edu.cn
{lijian83,zhiliny}@mail.tsinghua.edu.cn

## ABSTRACT

Recent work has achieved remarkable zero-shot performance with multi-task prompted pretraining, but little has been understood. For the first time, we show that training on a small number of key tasks beats using all the training tasks, while removing these key tasks substantially hurts performance. We also find that these key tasks are mostly question answering (QA) tasks. These novel findings combined deepen our understanding about zero-shot generalization—training on certain tasks such as QA encodes general knowledge transferable to a wide range of tasks. In addition, to automate this procedure, we devise a method that (1) identifies key training tasks without observing the test tasks by examining the pairwise generalization results and (2) resamples training tasks for better data distribution. Empirically, our approach achieves improved results across various model scales and tasks. [1]

## 1 INTRODUCTION

Recent work (Brown et al., 2020; Artetxe et al., 2022; Rae et al., 2021) has demonstrated the potential of leveraging pretrained language models (PLMs) to perform zero-shot generalization. Zero-shot generalization enables PLMs to adapt to a variety of natural language processing (NLP) tasks without relying on any annotated data, which opens the possibility towards generic systems.

Pretrained models, such as GPT-3 (Brown et al., 2020), BERT (Devlin et al., 2019) and T5 (Raffel et al., 2020), can perform zero-shot inference on unseen test tasks by leveraging natural language prompts and formulating NLP tasks into language modeling tasks. More recent advances (Wei et al., 2022; Sanh et al., 2022) performed **multi-task prompted training** on PLMs and further enhanced the zero-shot performance to a large extent. Despite the substantial progress, few works have studied how multi-task prompted training boosts the zero-shot performance. The lack of understanding hinders further improvement of the field.

To this end, we take a further step to understand multi-task training for zero-shot generalization (1) by selecting only a small number of key training tasks and performing multi-task training and (2) by studying the characteristics of tasks with general transfer ability. The results reveal several interesting findings — First of all, only a small number of training tasks dominate the performance of zero-shot generalization. In other words, using only these key training tasks to perform multi-task training leads to good results, while removing these key tasks would drastically hurt the zero-shot performance. Secondly, not all tasks are born equal, and some tasks show general transfer ability by providing widely useful knowledge. Moreover, key tasks with general transfer ability can be automatically detected using pairwise generalization results.

In addition, based on the findings, we propose an improved method, **task resampling**, which improves multi-task training for zero-shot generalization to a large extent. Task resampling first automatically identifies a set of key training tasks based on pairwise training and evaluation without peeking forward

---

[*]Corresponding Authors.
[1]Our code is released at https://github.com/zhouj8553/Improving-T0.

any test tasks, and then performs resampling by upsampling key tasks or downsampling non-critical tasks, as shown in Figure 3. In this way, we build a better mixture of multi-task training sets to highlight those key tasks with better general transfer ability. Experiments show that task resampling consistently outperforms the previous approach T0 (Sanh et al., 2022) across three different model scales and on test tasks of various types.

To sum up, our contributions are as follows.

1. We conduct experiments to understand and reveal how multi-task training for zero-shot generalization works—(1) Only a small number of training tasks dominate zero-shot generalization; (2) Some key tasks provide general transfer ability and can be detected using pairwise generalization results.

2. We devise a novel method, task resampling, to improve zero-shot generalization by (1) first automatically identifying key training tasks based on pairwise training and evaluation without observing any test tasks, and (2) resampling training tasks using upsampling and downsampling strategies.

3. Experiments show that our approach achieves new state-of-the-art results across various model scales and tasks.

## 2 RELATED WORK

### 2.1 ZERO-SHOT LEARNING IN NLP

Zero-Shot Learning denotes the setting when no data correlated with the test set is available during the training stage. The early definition of zero-shot learning referred to predicting samples with unseen classes, so traditional methods require prior information such as semantic knowledge (Zhang et al., 2019a) or knowledge graph (Chen et al., 2021) for an unseen class so that model can predict that class without training data. Meta-learning (Zhang et al., 2022) and reinforcement learning (Ye et al., 2020) methods are also used for zero-shot learning.

Recently, more work focuses on the setting of predicting samples with unseen tasks, supported by the development and prevalence of pre-trained language models (PLMs), as well as multi-task training. McCann et al. (2018) unifies NLP tasks into QA-format to perform multi-task learning. Liu et al. (2019) designs a multi-task deep neural network for natural language understanding tasks. Aghajanyan et al. (2021) designs an intermediate training stage between pretraining and finetuning using around 50 tasks. Most recently, with the combination of the above two approaches, T0 and FLAN (Sanh et al., 2022; Wei et al., 2022) have shown that explicit multi-task prompted training where all tasks are unified by the natural language prompts can vastly promote zero-shot task generalization. We build upon previous work within this new paradigm and devote ourselves to improving the recipe of multi-task prompted training by revealing the mechanism of generalization in Section 3 and further enhancing its performance in Section 4.

### 2.2 THE INTERPRETATION OF PROMPTED LEARNING

Recent work has shown an increased interest in how the prompts help the model generalize to unseen tasks. Some researchers (Wei et al., 2022; Schick & Schütze, 2021; Mishra et al., 2022) suggest that the model learns to understand what they are doing through prompts. While some work (Webson & Pavlick, 2022; Logan IV et al., 2022) challenges this assumption, revealing that sometimes we could get comparable performance without prompts or even with wrong prompts.

T0 (Sanh et al., 2022) claims that they only empirically witness the transfer phenomenon, but it is unclear why it happens. We provide new insights into the reason for generalization. We challenge the idea that the model learns the task through instructions, based on the observation that deleting a small but important set of tasks will lead to transfer failure. We suggest that most generalization ability comes from key tasks, which could be divided into specific and general transfer abilities. We hope our discovery could promote the development of this field.

## 2.3 Transfer Relationships in Multi-Task Learning

Based on the observation that transfer ability comes from key tasks, we would like to further improve the zero-shot performance by exploring the transfer relationships in multi-task learning. Some previous works learn transfer relationships in supervised multi-task training. Taskonomy (Zamir et al., 2018) builds a transfer structure of a series of computer vision tasks by training task-specific encoders on each dataset and retraining the decoders on target datasets. Dwivedi & Roig (2019) and Song et al. (2019) obtain the transfer relationship based on the assumption that transferable task-specific models have similar representations or embeddings on the same data. Vu et al. (2020) learns the embedding of each NLP task and tries to predict the transfer relationship between different datasets. ExT5 (Aribandi et al., 2022) learns the transferability by co-training the source task and target task and evaluate on the target task. UnifiedQA (Khashabi et al., 2020) explores the transferability between different QA tasks. The major challenge lies in that we could not get access to the test task at the training stage, thus it is hard to figure out in advance which training tasks can be more useful for unseen tasks. To address this challenge, we propose a reweighting method based on the transfer performance of pairwise training tasks, which will be discussed in Section 4.

## 2.4 Data Augmentation in NLP

Data Augmentation is widely used in NLP to strengthen the robustness and diversity of the data distribution and promote the model performance. However, most of the approaches are conducted in word-level (Zhang et al., 2015; Wang & Yang, 2015; Wei & Zou, 2019) and sentence-level (Kafle et al., 2017; Hou et al., 2018; Khashabi et al., 2018; Zhang et al., 2018), or in other words, instance-level. In our paper, to naturally cater for the multi-task setting, we adopt a new perspective and design a cross-domain data augmentation which intersects each domain data with different kinds of tasks and significantly improves the diversity of data distribution.

## 3 Understanding Zero-Shot Task Generalization

This section explores how multi-task training contributes to zero-shot generalization. By revealing the mechanism of task transfer, we provide some insights for improving zero-shot performance.

### 3.1 Data

We followed the setting of T0 (Sanh et al., 2022) and adopted the tasks therein. There are 38 training tasks across 8 task types, and 11 test tasks ranging from natural language inference (RTE (Candela et al., 2006), CB (De Marneffe et al., 2019), ANLI/R1-R3 (Nie et al., 2020)), coreference resolution (WSC (Levesque et al., 2012), Winogrande (Sakaguchi et al., 2020)), sentence completion (COPA (Roemmele et al., 2011), StoryCloze (Mostafazadeh et al., 2017), Hellaswag (Zellers et al., 2019)), to word disambiguation (WiC (Pilehvar & Camacho-Collados, 2019)). Both training and test sets are disjoint in task types, thus guaranteeing the zero-shot setting. We report the mean and median accuracy over multiple prompts for each test task.

### 3.2 A Small Number of Key Tasks Dominate Performance

Since there has been little agreement on where the excellent zero-shot generalization performance comes from, we would like to conduct experiments to explore its mechanism. Some researchers believe that the model understands the prompts through multi-task training (Wei et al., 2022; Schick & Schütze, 2021; Mishra et al., 2022), while we take an orthogonal perspective and hypothesize that there might be a few key tasks that are crucial for the zero-shot generalization performance. For a straightforward verification of the above hypothesis, we conduct two experiments.

**Single Task Shows Zero-Shot Transfer Ability**  We set up an experiment to study the pairwise transfer results between all pairs of tasks. Specifically, for any pair of tasks in the T0 collection, we train on one task and evaluate on the other. We expect to decouple the effects of multi-task learning and observe the transfer ability of single tasks. Results are shown in Figure 1. On the held-out test tasks, the performance gap between multi-task training and single-task training is less than 5 points on average. We also conduct experiments by training on top-3 tasks for each test dataset. The detailed results are presented in Appendix D.1.

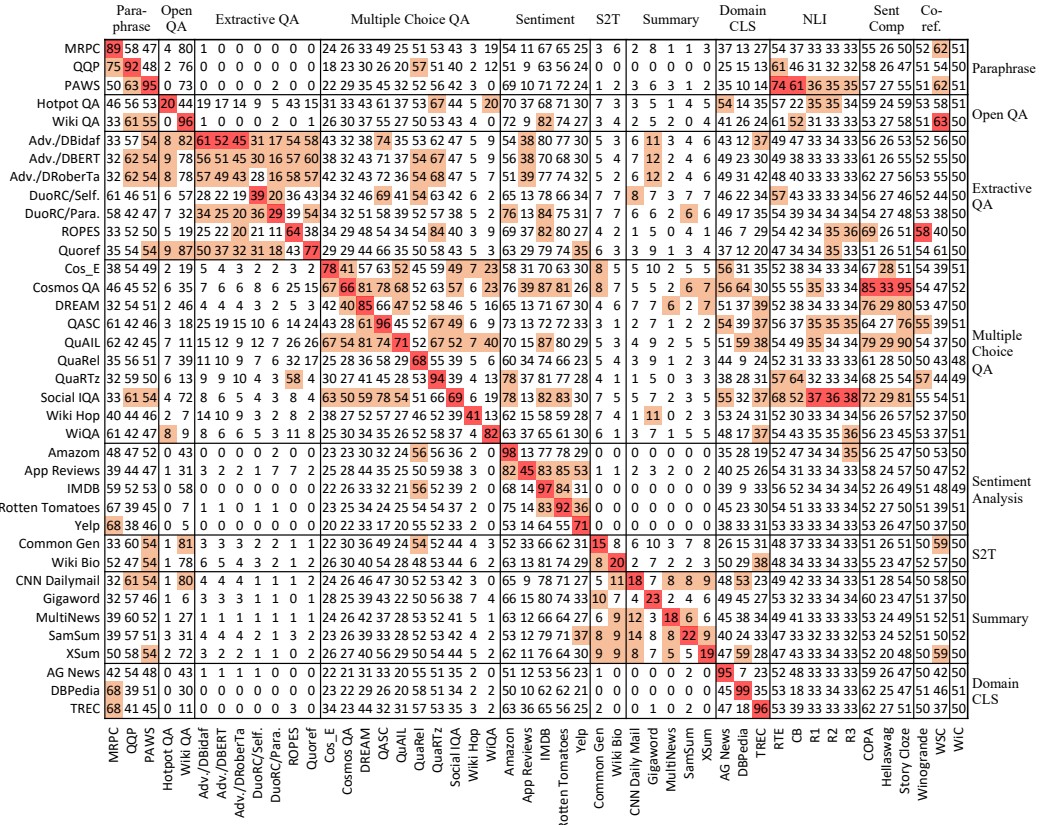

Figure 1: Pairwise transfer relationships on T5-XL. The entry at row $i$ and column $j$ denotes the average performance when the model is trained on task $i$ and evaluated on task $j$. For each entry, the value is the average score of different prompts. (Accuracy if only Accuracy is calculated, and otherwise the mean of Accuracy and F1.) Only those prompts related to the original tasks are included for evaluation. We highlight those entries with high scores for each task (Red is the Top-1). The horizontal and vertical lines denote the boundary of task-type groups.

**A Small Set of Tasks Dominates Performance on the Test Tasks** We manually select the top-8 tasks out of all training tasks of T0. These selected tasks empirically demonstrate good generalization to the test tasks according to preliminary experiments. Specifically, we first select the top-3 key tasks for each test task. Then, we select the tasks which appear at least twice in the top-3 support tasks. As a result, the number of selected key tasks is exactly 8. The top-8 selected tasks are CosmosQA (Huang et al., 2019), Social IQA (Sap et al., 2019), PAWS (Zhang et al., 2019b), QuAIL (Rogers et al., 2020), Wiki QA (Yang et al., 2015), QuaRTz (Tafjord et al., 2019), QASC (Khot et al., 2020), and ROPES (Lin et al., 2019).

For comparison, we experimented with three different variants by training a T5-Large model using only top-8 tasks ("Top-8 Only") and all but the top-8 tasks ("T0 Tasks w/o Top-8"), respectively. We also experimented using backbones with different scales (XL) or architectures (decoder-only model), and results are presented in Appendix C.

Results on T5-Large are shown in Table 1. We observe that training with only the top-8 tasks outperforms training with all tasks when tested on 11 test tasks, while training with all but the top-8 tasks drastically decreases performance. It proves that a few key tasks contribute to zero-shot task generalization. Training with key tasks selected by post-hoc results achieves much better zero-shot performance than training with all tasks. It is generally agreed that training with more tasks should lead to better learning of prompts, but our experiments show that this is not the key reason for the performance improvement. The information contained in the key tasks plays a significant role in improving the few-shot generalization. This result raises a further question whether the model learns to read, understand and react to instructions broadly from all tasks, or just benefits from several key tasks which have strong general transferability.

| Train Tasks | Met. | Natural Language Inference | | | | | Sentence Completion | | | Co-Reference | | WSD. | Avg. |
|---|---|---|---|---|---|---|---|---|---|---|---|---|---|
| | | RTE | CB | ANLI1 | ANLI2 | ANLI3 | COPA | Hella. | Story. | WSC | Wino. | WiC | |
| All T0 Tasks | Mean | 72.53 | 50.60 | 30.93 | 31.96 | 32.23 | 82.20 | 27.16 | 92.05 | 62.21 | 52.00 | 50.14 | 53.09 |
| | Med. | 74.01 | 57.14 | 30.40 | 31.60 | 31.75 | 83.00 | 27.60 | 91.77 | 62.98 | 52.33 | 50.00 | 54.87 |
| Top-8 Only | Mean | **73.10** | **66.55** | **33.55** | 32.46 | **36.34** | **84.62** | **30.93** | **94.51** | **64.04** | **53.02** | 50.52 | **56.33** |
| | Med. | **74.91** | **71.42** | **33.10** | 32.10 | **36.42** | **84.50** | **30.96** | **94.76** | **65.38** | **52.96** | 50.16 | **56.97** |
| T0 Tasks w/o Top-8 | Mean | 60.47 | 44.17 | 30.68 | **32.81** | 32.87 | 67.07 | 26.46 | 68.81 | 51.83 | 51.29 | **50.92** | 47.03 |
| | Med. | 60.29 | 44.64 | 30.80 | **33.00** | 33.33 | 66.83 | 26.65 | 72.53 | 47.12 | 51.54 | **50.71** | 47.04 |

Table 1: Zero-shot performance of training with/without top-8 tasks (out of 38) on T5-Large. The top-8 tasks are CosmosQA, Social IQA, PAWS, QuAIL, Wiki QA, QuaRTz, QASC, and ROPES. "Top-8 Only" means using only the top-8 tasks. "T0 Tasks w/o Top-8" means using the T0 tasks with top-8 tasks removed. Results that are comparable to or outperform the T0 baseline are denoted in bold.

## 3.3 General Transfer and Specific Transfer

First of all, we divide the transfer ability into specific transfer ability and general transfer ability according to the scope of target tasks.

**Specific transfer ability** means that the task can only provide special knowledge for a small set of tasks with certain kinds of patterns. A typical example is the sentiment analysis task, which helps the sentiment analysis tasks of different domains a lot, but has little effect on other complex NLU tasks. The specific transfer ability is relatively stronger among the tasks at the diagonal blocks in Figure 1.

**General transfer ability** means that the task can provide knowledge that is required by most downstream tasks. The more it provides beyond the knowledge captured by the pretrained model, the better it will contribute to the overall transfer ability of the model. For example, adding question answering (QA) tasks will improve the performance on most of the downstream tasks, which could be because they provide valuable commonsense knowledge and reasoning skills.

After introducing the division of transfer ability, we raise two questions based on the experimental results in Section 3.2. (1) Do the key tasks embrace the ability of general transfer or specific transfer? (2) Can we reveal the common patterns shared by the tasks with general transfer ability?

## 3.4 Not All Tasks are Born Equal

**Some QA Tasks Show General Transfer Ability** For the first question, from Figure 1, most of the tasks selected in post-hoc experiments bring improvements on a wide range of tasks, and thus have a certain degree of general transfer ability. In addition, most of the tasks that show general transfer ability are QA tasks.

Two notable concepts are QA format and QA tasks. QA tasks indeed take the QA format. However, QA-formatted data are not necessarily QA tasks (e.g., prompted sentiment analysis data taking the QA format is not a QA task.). Here, QA tasks refer to those tasks that require reasoning skills, such as reading comprehension. We conduct an experiment by formatting the sentiment analysis task into a multiple-choice QA task. Specifically, we convert the format of a sentiment analysis task: Yelp_Review_Full into the multiple choice QA format (see Figure 2). We compare the zero-shot performance on 11 unseen tasks between training on raw Yelp_Review_Full and training on QA-formatted Yelp_Review_Full, which is displayed in Table 2. Results show that simply using QA-formatted non-QA-tasks does not benefit zero-shot performance, proving that it is not simply the QA format that results in the zero-shot ability.

| Task | Met. | Natural Language Inference | | | | | Sentence Completion | | | Co-Reference | | WSD | Avg. |
|---|---|---|---|---|---|---|---|---|---|---|---|---|---|
| | | RTE | CB | ANLI1 | ANLI2 | ANLI3 | COPA | Hella. | Story. | WSC | Wino. | WiC | |
| Yelp | Mean | 52.71 | 32.50 | 33.38 | 33.59 | 33.38 | 53.30 | 25.65 | 46.69 | 36.54 | 49.93 | 49.97 | 40.69 |
| | Med. | 52.71 | 39.29 | 33.40 | 33.40 | 33.50 | 54.00 | 25.82 | 46.77 | 36.54 | 50.08 | 50.00 | 41.41 |
| Yelp2QA | Mean | 52.78 | 37.02 | 33.27 | 33.49 | 33.57 | 57.37 | 24.93 | 50.24 | 36.54 | 50.09 | 49.86 | 41.74 |
| | Med. | 52.71 | 41.07 | 33.40 | 33.40 | 33.50 | 58.00 | 24.96 | 50.45 | 36.54 | 50.28 | 50.00 | 42.21 |

Table 2: Zero-shot performance of T0-XL trained on Yelp_Review_Full and QA-formatted Yelp_Review_Full respectively.

Considering that QA tasks account for a large portion of the T0 benchmark, which might affect our conclusion, we also validated it in another setting (i.e., QA tasks only account for 7/43 of all training tasks) and presented the results in Appendix C.3. The results verify the effectiveness of QA tasks.

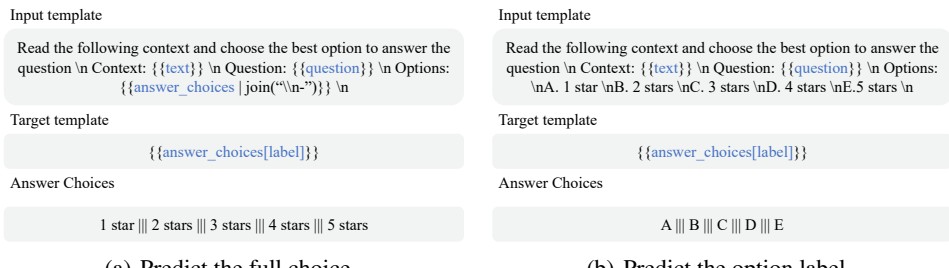

| Input template | Input template |
|---|---|
| Read the following context and choose the best option to answer the question \n Context: {{text}} \n Question: {{question}} \n Options: {{answer_choices | join("\\n-")}} \n | Read the following context and choose the best option to answer the question \n Context: {{text}} \n Question: {{question}} \n Options: \nA. 1 star \nB. 2 stars \nC. 3 stars \nD. 4 stars \nE.5 stars \n |
| Target template | Target template |
| {{answer_choices[label]}} | {{answer_choices[label]}} |
| Answer Choices | Answer Choices |
| 1 star ||| 2 stars ||| 3 stars ||| 4 stars ||| 5 stars | A ||| B ||| C ||| D ||| E |
| (a) Predict the full choice. | (b) Predict the option label. |

Figure 2: An illustration of multiple-choice QA formatted sentiment analysis prompts.

**Why QA Tasks Show General Transfer Ability?** Experiments show that some QA tasks demonstrate general transfer ability, and we would like to explore the underlying reasons. We suspect some QA tasks provide some knowledge that is not captured in the pretraining process. From the examples in Table 3, we can see that both CosmosQA and Social IQA require some simple reasoning ability in the general domain, which is required for a wide range of NLP tasks. More importantly, it is difficult to learn this knowledge in the pretraining stage, so an additional supplement is necessary to make the model have good cognitive ability. As a result, those tasks show better general transfer ability. There may be some quantitative methods to evaluate the knowledge provided by these tasks. A possible solution is to design some probe tasks, as is done in Pruksachatkun et al. (2020), and we leave this for future work.

| | |
|---|---|
| CosmosQA | **Context:** So , last day in Seattle , and my flight was at 1:30 . I got to chit chat with my old manager ( more like a mentor ) , and left Seattle feeling really good and inspired . . 
 **Question:** Why did I chit chat with my old manager ? 
 **Answer:** Because I enjoy talking to him . |
| Social IQA | **Context:** Cameron decided to have a barbecue and gathered her friends together. 
 **Question:** How would Others feel as a result? 
 **Answer:** like attending |
| WikiHop | **Question:** participant_of juan rossell 
 **Answer:** 1996 summer olympics |
| WiQA | **Paragraph Step Context:** [ "Plants and animals long ago died", "They are buried under layers of soil", "Pressure builds over time", "The remains liquefy", "The carbon atoms rearrange to become a new substance."] 
 **Question:** suppose a smaller satellites is determined happens, how will it affect less remains liquefy. 
 **Answer:** no effect |

Table 3: Examples of part of the QA tasks. CosmosQA and Social IQA both show excellect general transfer ability, while WikiHop QA and WiQA don't. The biggest difference between them is the knowledge domain.

**Which Kinds of QA Tasks Work?** Another important observation is that not all QA tasks show general transfer ability. Specifically, CosmosQA, Social IQA, and QuAIL show outstanding transfer ability, while some tasks such as WikiHop, and WiQA do not. Through a careful examination of the datasets (see Table 3 for reference), we conjecture that there are two reasons. First of all, the domain type matters a lot. Taking an extreme case as an example, training on datasets full of math problems is not likely to provide general transfer ability to other tasks. All of CosmosQA, Social IQA, and QuAIL require commonsense knowledge that is useful in the general domain. However, the WikiHop dataset urges the model to remember specific knowledge that is mainly required for knowledge contests. Another possible factor is the text format. In detail, the expressions of WikiHop/WiQA seem much more artificially-constructed than Social IQA/CosmosQA, thus showcase limited transfer ability.

So far, we have analyzed phenomena about the zero-shot task generalization ability. At the same time, three problems remain unsolved: (1) The test set cannot be seen in advance. (2) We need to further distinguish which QA tasks are useful when a large number of QA tasks are provided. (3) When the training set provided is changed, we need to distinguish new tasks with general transfer

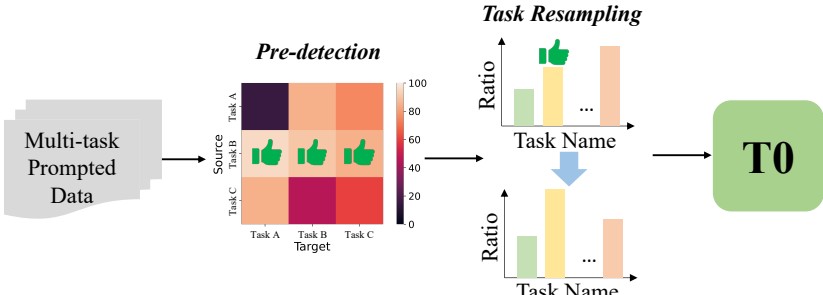

Figure 3: The pipeline of the improved multi-task prompted training recipe. We detect key tasks by examining the pairwise generalization results between training tasks. These key tasks are upsampled, or non-key tasks are downsampled to form an optimized mixture of datasets.

ability. For the above three reasons, we propose a general data-driven approach to identify tasks with general transfer capability.

## 4 AN IMPROVED METHOD: TASK RESAMPLING

### 4.1 METHOD

We have arrived at several findings in Section 3, revealing that (1) only a small number of training tasks dominate the performance of zero-shot generalization and (2) some key tasks with general transfer ability can be detected through pairwise generalization. Following the findings, we hypothesize that one of the key aspects of improving zero-shot performance is to appropriately adjust the weight of different training tasks, such that the model is trained with an optimized mixture of multi-task datasets. To this end, we devise a novel method, task resampling, that first automatically identifies a set of key training tasks based on pairwise training and evaluation without observing any test tasks, and then adjusts the training data distribution through upsampling or downsampling. Figure 3 shows an overview of our method.

Formally, we are given a set of training tasks $\mathcal{T} = \{t_i\}_i$ where $t_i$ is a task, and a pretrained model $\mathcal{M}$. Each task is formulated as $t_i = \{x_{i,j}, y_{i,j}\}_j$, consisting of a prompted input $x_{i,j}$ and a prompted target $y_{i,j}$. Our goal is to assign appropriate weights $\{w_i\}_i$ for each training task, and use the optimized mixture of training tasks to train the model $\mathcal{M}$ in a multi-task manner such that it performs well on unseen test tasks.

Generally, our method consists of three major steps.

1. **Pre-detection of key tasks**. Identify key training tasks based on pairwise training and evaluation without relying on any test tasks. This can be viewed as a prior approximation to the post-hoc method in Section 3.2.

2. **Task resampling**. Resample different training tasks by upsampling key tasks or downsampling non-key tasks.

3. **Multi-task training**. Train a model using the resampled mixture of multi-task datasets.

**Pre-detection of Key Tasks**  Without observing the test tasks, the main idea of our approach is to use pairwise training and evaluation within the training tasks. To identify the key tasks, we first train a model on each training task and evaluate it on all the training tasks. This results in an $N \times N$ ($N$ is the number of training tasks) table, which is part of the results in Figure 1. Then we design a method to select key training tasks based on this $N \times N$ table. For each task pair $A$ and $B$, let $f(A, B)$ be the performance of training on $A$ and evaluating on $B$. We let $g(A, B) = 1$ if the following conditions are satisfied (and otherwise $g(A, B) = 0$):

1. $A$ and $B$ are of different task types.

2. The performance $f(A, B)$ is high enough:

$$f(A, B) \geq \max_{A' \neq B} f(A', B) - \text{TH}_1 \qquad \text{and} \qquad f(A, B) \geq \text{mean}_{A' \neq B} f(A', B) + \text{TH}_2.$$

Here $TH_1$ and $TH_2$ are two constants which control the tolerant distance between f(A, B) and maximum performance and average performance respectively. The value $g(A, B)$ indicates whether $A$ is a high-performing training task for $B$. We constrain that $A$ and $B$ are of different types because we eventually target cross task generalization. We then aggregate $g(A) = \sum_B g(A, B)$ to represent how many times $A$ is a high-performing training task for another task, and use the tasks with the largest $g(A)$ values as the key tasks. In our implementation, we apply thresholding on $g(A)$ to obtain the set of key tasks.

**Task Resampling by Upsampling or Downsampling**   After detecting the key tasks using only the training tasks, we propose a simple yet effective resampling approach to optimize the mixture of multi-task data. We either perform upsampling or downsampling strategy. For upsampling, we upsample the key tasks by $N_u$ times. For downsampling, we cap the number of samples to be $N_d$ for each non-key task and use the original sample size for the key tasks. In our preliminary experiments, we found task resampling more robust than using the key tasks only because it takes a softer approach to highlight the importance of key tasks while maintaining knowledge from other tasks.

**Data Augmentation**   In the above sections, we have discussed an important discovery that certain tasks are crucial for zero-shot performance. However, some of these key tasks might be limited in terms of labeled data. Thus, we further propose a data augmentation method to create as many samples as possible for each task. Specifically, given tasks $A$ and $B$, we apply the prompts of $A$ to the data of $B$ to obtain additional augmented data. In other words, we have more data from task $B$ that are used to perform the task $A$. We use a trained T0 to predict the labels of the augmented samples, which is similar to self-training. Note that data augmentation is optional and independent of task resampling. We do ablation studies to investigate the effectiveness of this component.

**Multi-task Training**   Our training procedure is the same as T0. The only difference is that we employ an optimized mixture of datasets (and optionally with data augmentation).

## 4.2   EXPERIMENTAL SETUP

Following the same training and evaluation setting as T0 (Sanh et al., 2022), we finetune the T5-LM-Adapt model on 38 training tasks, which has been discussed in detail in Section 3.1. For data preprocessing, following T0, to balance the number of data for different tasks, we restrict the maximum number of data examples for each training task to 500,000.

Based on our resampling strategy, we set the pre-detection parameters as $TH_1 = 5$, $TH_2 = 10$, and then choose the datasets which are counted as the key tasks at least twice (i.e., all tasks $A$ with $g(A) \geq 2$). Given each key task $D$ with data size $|D|$, we duplicate $D$ by 5 times ($N_u = 5$) for the upsampling strategy and empirically start from 50,000 samples for each dataset. For the downsampling strategy, we downsample each non-key task to $N_d = \min(50,000, |D|)$ samples. We provide detailed statistics about the datasets in Appendix B.1.

## 4.3   RESULTS

**Post-hoc v.s. Prior Detection of Key Tasks**   One vital part of our approach is the detection of key tasks, so we list the key tasks selected by post-hoc (i.e., observing the test tasks) and pre-detection methods in Table 4. There are five common tasks shared by the two methods, indicating that our approach can detect most of the key tasks. Moreover, even though the two sets of key tasks are not exactly matched, our experiments demonstrate that this does not affect performance.

| Method | Key Tasks |
|---|---|
| Post-hoc | Cosmos QA, Social IQA, PAWS, QuAIL, Wiki QA, QuaRTz, QASC, ROPES |
| Prior-detect | Cosmos QA, Adv./DBiDAF, Adv./DRoBERTa, QuaRTz, Social IQA, Hotpot QA, Adv./DBERT, ROPES, QuAIL |

Table 4: Key tasks selected by post-hoc and prior detection method. The underlined text represents the key tasks shared by both approaches.

**Main Results**   We experiment with three scales: Large, XL, and XXL. We compare our proposed method with the original T0 reported in (Sanh et al., 2022) (T0 (†)) and our reproduced T0 (T0 (*)). Also, we present the results of task-resampled T0, including T0 with upsampling key tasks (US-T0)

and T0 with downsampling non-key tasks (DS-T0). Besides, we display the performance of the task-resampled T0 with augmented data, dubbed as US+DA-T0 and DS+DA-T0, respectively. We summarize the following key observations from Table 5.

| Model | Met. | Natural Language Inference | | | | | Sentence Completion | | | Co-Reference | | WSD. | Avg. |
|---|---|---|---|---|---|---|---|---|---|---|---|---|---|
| | | RTE | CB | ANLI1 | ANLI2 | ANLI3 | COPA | Hella. | Story. | WSC | Wino. | WiC | |
| | | | | | | T5-Large-LM-Adapt (770M) | | | | | | | |
| T0 (*) | Mean | 72.53 | 50.60 | 30.93 | 31.96 | 32.23 | 82.20 | 27.16 | 92.05 | 62.21 | 52.00 | 50.14 | 53.09 |
| | Med. | 74.01 | 57.14 | 30.40 | 31.60 | 31.75 | 83.00 | 27.60 | 91.77 | 62.98 | 52.33 | 50.00 | 54.87 |
| DS-T0 | Mean | 74.22 | 60.95 | 35.65 | 32.57 | 35.88 | 87.64 | **28.29** | 94.12 | **63.75** | 54.89 | **51.60** | 56.33 |
| | Med. | 75.45 | 66.07 | 36.00 | 32.40 | 36.50 | 87.50 | **28.38** | 94.01 | 64.42 | 55.33 | **51.41** | 57.04 |
| DS+DA-T0 | Mean | **80.72** | **71.90** | 36.00 | **34.80** | 38.18 | 84.10 | 26.00 | 94.00 | 63.27 | 54.54 | 50.58 | 57.65 |
| | Med. | **81.23** | **80.36** | 36.40 | **35.20** | 39.33 | 85.21 | 26.06 | 94.39 | 63.94 | 54.38 | 50.31 | 58.80 |
| US-T0 | Mean | 78.30 | 61.55 | 36.05 | 34.06 | 36.50 | 87.79 | 28.05 | **94.97** | 62.40 | **55.52** | 51.46 | 56.97 |
| | Med. | 79.00 | 69.60 | 36.10 | 33.90 | 36.75 | 89.00 | 28.14 | **94.92** | 64.42 | **56.43** | 50.39 | 58.06 |
| US+DA-T0 | Mean | 80.69 | 70.95 | **37.38** | 34.20 | **39.43** | **87.97** | 26.73 | 93.71 | 63.27 | 55.58 | 51.47 | **58.31** |
| | Med. | 80.69 | 80.35 | **38.00** | 34.20 | **40.33** | **89.29** | 26.98 | 93.91 | 64.42 | 55.72 | 51.25 | **59.56** |
| | | | | | | T5-XL-LM-Adapt (3B) | | | | | | | |
| T0 (†) | Mean | 64.55 | 45.36 | 33.84 | 33.11 | 33.33 | 72.40 | 27.29 | 84.03 | 65.10 | 50.97 | 50.69 | 50.97 |
| | Med. | 64.08 | 50.00 | 33.65 | 33.40 | 33.33 | 74.92 | 27.51 | 85.09 | 64.42 | 50.51 | 50.39 | 51.57 |
| T0 (*) | Mean | 80.72 | 67.62 | 41.09 | 37.79 | 40.38 | 91.92 | **32.03** | 97.27 | 65.96 | 57.84 | 50.14 | 60.37 |
| | Med. | 80.14 | 75.00 | 42.80 | 39.20 | 41.75 | 92.00 | **32.29** | 97.22 | 68.27 | 58.41 | 50.00 | 61.62 |
| DS-T0 | Mean | 83.21 | 73.33 | 44.38 | 38.84 | 43.72 | 94.17 | 31.21 | **97.72** | 64.42 | 62.67 | 52.01 | 62.34 |
| | Med. | 82.67 | 82.14 | 45.40 | 39.70 | 45.58 | 94.50 | 32.03 | **97.70** | 64.42 | 63.38 | 51.33 | 63.53 |
| DS+DA-T0 | Mean | **84.77** | 74.40 | 43.25 | **39.17** | 43.22 | **94.93** | 27.01 | 97.65 | 62.02 | **66.74** | 53.09 | **62.39** |
| | Med. | **84.66** | 82.14 | 46.30 | **39.70** | 45.75 | **95.00** | 27.00 | 97.65 | 62.98 | 65.35 | 52.90 | **63.58** |
| US-T0 | Mean | 82.41 | 69.38 | 43.20 | 38.40 | 40.72 | 93.55 | 30.30 | 97.25 | 61.41 | 60.56 | **53.66** | 60.99 |
| | Med. | 82.34 | 82.81 | 45.41 | 39.94 | 42.60 | 93.75 | 30.38 | 97.34 | 63.67 | 62.27 | **52.66** | 63.02 |
| US+DA-T0 | Mean | 83.29 | **75.83** | **44.80** | 39.09 | **43.68** | 94.81 | 26.29 | 96.94 | 61.73 | 66.03 | 53.28 | 62.34 |
| | Med. | 84.48 | **82.14** | **47.90** | 39.90 | **47.25** | 94.50 | 26.17 | 97.06 | 64.90 | 65.11 | 52.98 | 63.85 |
| | | | | | | T5-XXL-LM-Adapt (11B) | | | | | | | |
| T0 (†) | Mean | 80.83 | 70.12 | 43.56 | 38.68 | 41.26 | 90.02 | 33.58 | 92.40 | 61.45 | 59.94 | 56.58 | 60.77 |
| | Med. | 81.23 | 78.57 | 44.70 | 39.40 | 42.42 | 90.79 | 33.65 | 94.71 | 64.42 | 60.46 | 57.21 | 62.51 |
| T0 (*) | Mean | 84.01 | **72.26** | 47.89 | 42.80 | 46.49 | 91.60 | **35.27** | **98.15** | 62.69 | 69.46 | 54.83 | 63.98 |
| | Med. | 85.02 | **83.93** | 49.00 | 44.00 | 48.58 | 95.00 | **34.62** | **98.24** | 66.35 | 70.24 | 52.35 | 66.12 |
| Our Best | Mean | **85.56** | 72.50 | **48.28** | **43.81** | **47.62** | **95.18** | 27.91 | 97.46 | **67.02** | **71.41** | **56.29** | **64.82** |
| | Med. | **85.74** | 82.14 | **51.60** | **46.40** | **51.25** | **95.00** | 27.71 | 97.54 | 66.35 | **71.19** | **58.46** | **66.67** |

Table 5: Zero-shot performance for our improved T0 and original T0 at three different scales. Results with † are reported by Sanh et al., and results with ⋆ are reproduced in our experiments. US-T0 means T0 with upsampling key tasks, DS-T0 means T0 with downsampling non-key tasks, and DS+DA-T0 / US+DA-T0 represents DS-T0 / US-T0 with augmented data. "Our Best" is achieved with the US+DA-T0 setup.

1. Advantage of Task Resampling. Our task-resampled T0 with both upsampling and downsampling strategies (US-T0 and DS-T0) boosts the performance of reproduced T0. Specifically, DS-T0 outperforms T0 (*) by 3.2% at Large scale and 2.0% at XL scale, and US-T0 outperforms T0 (*) by 3.9% at Large scale and 0.6% at XL scale.

2. Advantage of Data Augmentation. Task-resampled T0 achieves better performance with augmented data. Specifically, downsampling T0 (DS-T0) increases by 1.3% with augmented data at Large scale and upsampling T0 (US-T0) increases by 1.4% with augmented data at XL scale. And T0 with both task resampling and augmented data achieves 0.8% gain with reproduced T0. It indicates that data augmentation can further strengthen the mixture of multi-task data.

3. Advantage of Our Implementation Framework. Our reproduced T0 result is better than the reported T0 (Sanh et al., 2022) by 9.4% at XL scale and 3.2% at XXL scale.

## 5 CONCLUSIONS

This work studies the principles of zero-shot generalization through pairwise experiments, and reveals that a small number of training tasks dominate performance. We further divide the transfer relationship into specific transfer and general transfer, and find that adding those tasks with general transfer ability will contribute to the performance gain for most tasks. Moreover, those tasks with general transfer ability can be identified by examining the pairwise generalization results. Based on the findings, we propose the task resampling method to improve the zero-shot performance. Extensive experiments demonstrate the effectiveness of our framework.

ACKNOWLEDGMENTS

Jian Li and Jing Zhou are supported in part by the National Natural Science Foundation of China Grant 62161146004, Turing AI Institute of Nanjing and Xi'an Institute for Interdisciplinary Information Core Technology.

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

# A EXPERIMENTAL DETAILS

## A.1 HYPER-PARAMETERS SELECTION

For all the experiments, we adopt the ADAM optimizer and use a learning rate of 1e-4. Considering the amount of data, the batch size and training steps are different for different settings in our experiments. We don't search the training hyper-parameters anymore, because we find that the performance is similar as long as we train for sufficient epochs in our preliminary experiments. We use different batch sizes and training steps for different amounts of data for time-saving. The hyper-parameters of our experiments are in Table 6.

| Experiment | Batch Size | Steps |
|---|---|---|
| Single Task Transfer | 512 | 1000 |
| Top-3 Task Transfer | 1024 | 2000 |
| Top-8 Task Transfer | 1024 | 10000 |
| Full Dataset | 1024 | 20000 |

Table 6: Training hyper-parameters for our experiments.

For other hyper-parameters we selected, we predefine them using some preliminary experiments on T5-Large, and then apply them directly on larger models for the sake of time. We first select the key tasks by searching $TH_1, TH_2$ using the upsampling strategy with $N_u = 5$, the whole search space is $TH_1 = \{5\}, TH_2 = \{5, 10\}$. We choose $G(A) \geq 2$ because if you draw the distribution of G(A) values, you can clearly see that the tasks with $G(A) \leq 1$ are the long-tailed part. When the key tasks are selected, we then search the hyper-parameters for upsampling and downsampling using the search space $N_u = \{2, 5\}, N_d = \{5, 10\}$, and choose the best one on T5-Large. We find that those hyper-parameters don't affect the results a lot, i.e., the gap among them is much smaller than the gap between them and the baseline.

We only report the best result on T5-XXL, because we are unable to run all the experiments due to the limitation of computing resources. Our best result is achieved using US+DA-T0.

## A.2 DATA AUGMENTATION DETAILS

We propose two algorithms for the domain-task intersection. The first one is based on a human-written taxonomy tree, and another relies on universal fields. We combine the two techniques and achieve a balance between quality and diversity.

**Taxonomy Tree Based Domain-Task Intersection**    To build a general taxonomy tree to cover as many prompts as possible, we take both task format and task content into consideration to develop a series of guidelines similar to a Decision Tree. Then, at the end of each branch, we intersect the source data lying in that branch with the related prompts belonging to that branch. In this way, we might produce a reasonable combination of the source data and prompt. For example, classification tasks like IMDB can also do tasks like title generation.

**Universal Domain-Task Generation**    To further improve the diversity of the augmented data, we get rid of the man-made restriction and propose universal domain-task generation. In detail, we define the unified fields for data from all domains, which are utilized for various tasks. Obviously, each original domain data lacks certain kinds of fields, e.g., AG NEWS data only have two fields: category label and text. Therefore, we leverage the T0 to predict the missing fields in order to conduct different kinds of tasks using the prompts having already been trained in T0. For some tasks, we also train a specific model for prediction to get better performance. After that, we filter samples according to the confidence score (i.e., the probability output by the model).

## B    MORE EXPLORATIONS ON THE TRANSFER ABILITY

### B.1    STATISTICS OF THE CURRECT DATASETS

We report the size of the dataset in our experiments in Table 7. To explore more statistical features of the original dataset, we further provide some statistical indicators. Specifically, we calculate the average sequence length (SLEN) and the mean segmental TTR (MSTTR) of the prompted input of training datasets. MSTTR is calculated with a window size of 50. We calculate those statistical values using at most 10000 examples for each prompted task for the sake of time. Results are in Table 7.

An interesting phenomenon is that datasets with fewer training examples are more likely to show general transfer ability. We speculate that it is because a lot of manual effort is needed to make up those datasets with sufficient knowledge. It is too costly to make these datasets extremely large. It seems that the statistics such as MSTTR or data length alone cannot be good indicators for transfer ability. More explorations about the influence factors of transfer ability are left for future work.

## C    MORE RESULTS ON OTHER MODELS AND OTHER BASELINES

### C.1    RESULTS ON LARGER MODEL

To verify whether the observation that a small number of key tasks dominate zero-shot performance still holds true on larger models, we conduct the top-8 tasks experiments on T5-XL, similar to Section 3.2. Results are in Table 8. From the results, we can see that the model trained on top-8 only slightly outperforms the baseline, while greatly defeating the performance of the model trained on all T0 tasks without the top-8 tasks.

### C.2    RESULTS ON DECODER-ONLY ARCHITECTURE

To verify whether the results still hold true on more architectures, we experiment on more architectures. Considering that the encoder-only model is not suitable for language modeling tasks, here we only consider the decoder-only architecture. In specific, we conduct the top-8 experiments on GPT-Neo-1.3B (Black et al., 2021). Results are in Table 9. From the results, we can see that, the observation that a small number of key tasks dominate zero-shot performance mentioned in Section 3.2 still holds true. We train with a prefix-LM loss.

| Task Name | Orig Num | T0 Num | US Num | DS Num | MSTTR | In Len | Out Len |
|---|---|---|---|---|---|---|---|
| MRPC | 3668 | 23288 | 23288 | 23288 | 0.74 | 53 | 1 |
| QQP | 363846 | 2183076 | 49998 | 50000 | 0.67 | 30 | 1 |
| PAWS | 49401 | 565240 | 565240 | 50000 | 0.64 | 47 | 1 |
| Hotpot QA | 88869 | 444345 | 250000 | 444345 | 0.75 | 18 | 2 |
| Wiki QA | 20360 | 108040 | 108040 | 50000 | 0.69 | 9 | 27 |
| Adv./DBidaf | 10000 | 50000 | 250000 | 50000 | 0.78 | 130 | 4 |
| Adv./DBERT | 10000 | 50000 | 250000 | 50000 | 0.78 | 146 | 4 |
| Adv./DRoberTa | 10000 | 50000 | 250000 | 50000 | 0.78 | 143 | 4 |
| DuoRC/Self. | 60721 | 545235 | 49995 | 50000 | 0.79 | 372 | 2 |
| DuoRC/Para. | 69524 | 604172 | 49995 | 50000 | 0.79 | 356 | 7 |
| ROPES | 10924 | 131088 | 655440 | 131088 | 0.76 | 211 | 1 |
| Quoref | 19399 | 213389 | 213389 | 50000 | 0.80 | 328 | 2 |
| Cos_E | 9741 | 107151 | 107151 | 50000 | 0.74 | 30 | 1 |
| Cosmos QA | 25262 | 328406 | 1642030 | 328406 | 0.76 | 10 | 8 |
| DREAM | 6116 | 30580 | 30580 | 30580 | 0.73 | 141 | 4 |
| QASC | 8134 | 65072 | 65072 | 50000 | 0.66 | 63 | 2 |
| QuAIL | 10246 | 133198 | 665990 | 133198 | 0.80 | 372 | 5 |
| QuaRel | 1941 | 9705 | 9705 | 9705 | 0.72 | 57 | 2 |
| QuaRTz | 2696 | 21568 | 107840 | 21568 | 0.70 | 47 | 1 |
| SciQ | 11679 | 58395 | 58395 | 50000 | 0.73 | 89 | 2 |
| Social IQA | 33410 | 200460 | 1002300 | 200460 | 0.73 | 49 | 1 |
| Wiki Hop | 43738 | 393642 | 393642 | 50000 | 0.75 | 1248 | 2 |
| WiQA | 29808 | 238464 | 238464 | 50000 | 0.68 | 94 | 1 |
| Amazon | 3600000 | 499995 | 49995 | 50000 | 0.79 | 95 | 1 |
| App Reviews | 288065 | 1152260 | 50000 | 50000 | 0.56 | 33 | 1 |
| IMDB | 25000 | 275000 | 275000 | 50000 | 0.81 | 209 | 4 |
| Rotten Tomatoes | 8530 | 85300 | 85300 | 50000 | 0.73 | 29 | 1 |
| Yelp | 650000 | 499996 | 49994 | 50000 | 0.80 | 132 | 2 |
| Common Gen | 67389 | 606501 | 49995 | 50000 | 0.46 | 16 | 11 |
| Wiki Bio | 582659 | 500000 | 50000 | 50000 | 0.80 | 107 | 82 |
| CNN Daily Mail | 287113 | 2584017 | 49995 | 50000 | 0.83 | 66 | 338 |
| Gigaword | 3803957 | 499995 | 49995 | 50000 | 0.81 | 17 | 31 |
| MultiNews | 44972 | 269832 | 269832 | 50000 | 0.82 | 769 | 216 |
| SamSum | 14732 | 103124 | 103124 | 50000 | 0.78 | 98 | 20 |
| XSum | 204045 | 2040450 | 50000 | 50000 | 0.82 | 257 | 21 |
| AG News | 120000 | 840000 | 49994 | 50000 | 0.84 | 45 | 2 |
| DBPedia | 560000 | 500000 | 50000 | 50000 | 0.80 | 36 | 1 |
| TREC | 5452 | 47818 | 47818 | 47818 | 0.58 | 20 | 1 |

Table 7: Statistics of the training sets. "Orig Num" denotes the size of the original dataset. "T0 Num" denotes the size of prompted data in the T0 baseline. "US Num" denotes the size of prompted data we used in our upsampling experiments. "DS Num" denotes the size of prompted data we used in our downsampling experiments. "MSTTR" denotes the mean segmental TTR, which is an indicator to reflect lexical diversity. "In Len" denotes the average length of prompted input data. "Out Len" denotes the average length of target prompted target data.

| Train Tasks | Met. | Natural Language Inference | | | | | Sentence Completion | | | Co-Reference | | WSD | |
|---|---|---|---|---|---|---|---|---|---|---|---|---|---|
| | | RTE | CB | ANLI1 | ANLI2 | ANLI3 | COPA | Hella. | Story. | WSC | Wino. | WiC | Avg. |
| Baseline T0 | Mean | 80.72 | 67.62 | 41.09 | 37.79 | 40.38 | 91.92 | 32.03 | 97.27 | 65.96 | 57.84 | 50.14 | 60.37 |
| | Med. | 80.14 | 75.00 | 42.80 | 39.20 | 41.75 | 92.00 | 32.29 | 97.22 | 68.27 | 58.41 | 50.00 | 61.62 |
| Top-8 Only | Mean | 81.37 | 75.36 | 41.66 | 37.40 | 42.26 | 93.22 | 33.95 | 96.93 | 59.64 | 64.81 | 54.01 | 61.87 |
| | Med. | 81.59 | 73.21 | 41.70 | 37.50 | 43.50 | 93.00 | 33.36 | 96.90 | 60.06 | 65.87 | 54.94 | 61.97 |
| T0 Tasks w/o Top-8 | Mean | 55.45 | 45.24 | 34.60 | 34.05 | 34.81 | 84.68 | 28.73 | 86.50 | 54.29 | 42.50 | 55.94 | 50.62 |
| | Med. | 54.33 | 46.43 | 33.80 | 33.90 | 34.33 | 85.00 | 29.12 | 90.11 | 54.14 | 37.02 | 55.72 | 50.35 |

Table 8: Zero-shot performance of training with/without top-8 tasks (out of 38) on T5-XL. The top-8 tasks are CosmosQA, SocialIQA, PAWS, QuAIL, Wiki QA, QuaRTz, QASC, and ROPES. "Top-8 Only" means using only the top-8 tasks. "T0 Tasks w/o Top-8" means using the T0 tasks with top-8 tasks removed.

| Train Tasks | Met. | Natural Language Inference | | | | | Sentence Completion | | | Co-Reference | | WSD | |
| | | RTE | CB | ANLI1 | ANLI2 | ANLI3 | COPA | Hella. | Story. | WSC | Wino. | WiC | Avg. |
|---|---|---|---|---|---|---|---|---|---|---|---|---|---|
| Baseline T0 | Mean | 49.28 | 42.74 | 32.95 | 33.25 | 32.98 | 60.59 | 25.93 | 66.05 | 62.31 | 49.76 | 50.24 | 46.01 |
| | Med. | 47.83 | 50.00 | 33.20 | 33.30 | 33.00 | 60.02 | 26.07 | 68.36 | 63.46 | 49.49 | 50.16 | 46.81 |
| Top-8 Only | Mean | 62.92 | 61.07 | 30.24 | 32.07 | 32.18 | 68.20 | 26.06 | 71.23 | 58.17 | 49.41 | 51.27 | 49.35 |
| | Med. | 62.64 | 66.07 | 29.30 | 32.00 | 32.17 | 70.00 | 26.26 | 72.96 | 61.06 | 49.41 | 50.94 | 50.25 |
| T0 Tasks w/o Top-8 | Mean | 56.46 | 43.93 | 32.93 | 33.26 | 33.40 | 57.92 | 25.72 | 52.04 | 53.85 | 48.97 | 50.74 | 44.47 |
| | Med. | 55.78 | 50.00 | 32.90 | 33.30 | 33.17 | 57.50 | 25.70 | 52.70 | 57.69 | 48.86 | 50.63 | 45.29 |

Table 9: Zero-shot performance of training with/without top-8 tasks (out of 38) on GPT-Neo. The top-8 tasks are CosmosQA, SocialIQA, PAWS, QuAIL, Wiki QA, QuaRTz, QASC, and ROPES. "Top-8 Only" means using only the top-8 tasks. "T0 Tasks w/o Top-8" means using the T0 tasks with top-8 tasks removed.

## C.3 Results on Other Baselines

Since most tasks in T0 are defined as QA tasks, the observation that QA tasks are important might not be fair enough. Thus, we want to investigate whether the statement that "some general transfer classes dominate the zero-shot performance" still holds with a totally different mixture of datasets. Therefore, we consider conducting the experiment on the prompted datasets of FLAN (Wei et al., 2022). Noted that reading comprehension is one of the task types used in FLAN, which can be regarded as narrative QA (in this way, we classify QA tasks based on the content rather than the format), so we hope to explore what will happen if we remove all reading comprehension tasks in the mixture of datasets used in FLAN.

We use exactly the same datasets and prompts as FLAN (Wei et al., 2022), except that we include three dialogue datasets in the same way as in FLAN-T5 (Chung et al., 2022), and exclude the translation datasets. We leave the NLI and Commonsense Reasoning as the hold-out test set, which follows FLAN.

We conduct the experiments as follows: 1. Training with all remaining tasks in FLAN. (43 tasks in total); 2. Training with only the reading comprehension tasks in FLAN. (7 tasks in total); 3. Training without the reading comprehension tasks in FLAN. (36 tasks in total)

As can be seen in Table 10, the model which is trained on reading comprehension tasks greatly outperforms the model trained on FLAN tasks without reading comprehension tasks. Therefore, these experiments serve as supplementary to our main experiment conducted on T0 datasets.

| | Met. | Natural Language Inference | | | | | | | | | Sentence Completion | | | | Co-Reference | | |
| | | RTE | CB | ANLI1 | ANLI2 | ANLI3 | SNLI | MNLI | WNLI | QNLI | COPA | Hella. | PiQA | Story. | ARC/E. | ARC/C. | Avg. |
|---|---|---|---|---|---|---|---|---|---|---|---|---|---|---|---|---|---|
| Baseline | Mean | 73.24 | 78.57 | 42.08 | 40.08 | 43.17 | 58.33 | 61.04 | 52.54 | 74.58 | 82.88 | 41.42 | 67.38 | 90.61 | 58.39 | 40.58 | 60.33 |
| | Med. | 77.26 | 82.14 | 42.30 | 40.00 | 43.67 | 58.55 | 61.03 | 52.82 | 76.97 | 83.00 | 41.46 | 67.66 | 90.30 | 58.60 | 40.80 | 61.10 |
| RC. Only | Mean | 70.88 | 66.07 | 38.44 | 37.47 | 42.13 | 58.89 | 53.88 | 46.90 | 62.29 | 90.92 | 38.94 | 67.92 | 90.92 | 60.06 | 43.59 | 56.91 |
| | Med. | 75.81 | 67.86 | 38.8 | 37.40 | 42.42 | 59.23 | 58.58 | 46.48 | 65.73 | 90.70 | 38.99 | 68.25 | 90.70 | 60.18 | 43.31 | 57.95 |
| w/o RC. | Mean | 65.98 | 71.03 | 36.52 | 36.29 | 39.07 | 51.15 | 56.63 | 50.70 | 63.13 | 62.75 | 28.28 | 55.51 | 50.68 | 41.78 | 27.65 | 49.14 |
| | Med. | 70.40 | 71.43 | 36.00 | 36.40 | 39.25 | 50.04 | 57.09 | 50.70 | 64.54 | 62.00 | 28.41 | 55.58 | 49.65 | 41.67 | 27.76 | 49.39 |

Table 10: Results of training on FLAN. We train a T5-Large model using the datasets of FLAN. "RC" denotes Reading Comprehension. "ARC/E." denotes "ARC/Easy", and "ARC/C." denotes "ARC/Challenge".

# D Full Results

## D.1 Full Results on Top-3 key datasets

The results when training on the top-3 key datasets for each test task are in Table 11 and Table 12. We can see that the model trained on top-3 key datasets shows comparable results with the T0 baseline.

## D.2 Full Results Evaluated on Held-Out Test Datasets

Full results evaluated on held-out test datasets are in Table 13 and Table 14.

| Tasks | Met. | Natural Language Inference | | | | | Sentence Completion | | | Co-Reference | | WSD |
|---|---|---|---|---|---|---|---|---|---|---|---|---|
| | | RTE | CB | ANLI1 | ANLI2 | ANLI3 | COPA | Hella. | Story. | WSC | Wino. | WiC |
| Baseline T0 | Mean | 71.94 | 56.46 | 32.81 | 32.29 | 34.24 | 84.77 | 27.09 | 93.45 | 64.30 | 54.33 | 50.45 |
| | Med. | 71.88 | 60.94 | 32.32 | 31.64 | 34.05 | 85.54 | 26.92 | 93.23 | 65.23 | 54.38 | 50.31 |
| RTE_KEY | Mean | **72.42** | 54.17 | 32.08 | 31.87 | 34.22 | 69.41 | 28.37 | 82.43 | **64.52** | 51.78 | 50.67 |
| | Med. | **73.47** | 66.07 | 31.90 | 31.60 | 34.00 | 68.03 | 28.03 | 84.34 | **63.46** | 51.93 | 50.55 |
| CB_KEY | Mean | 56.86 | **60.48** | **33.17** | 32.94 | 34.98 | **81.04** | 28.48 | 93.34 | 44.81 | 54.57 | 51.79 |
| | Med. | 57.76 | **69.64** | **32.80** | 33.00 | 35.42 | **83.00** | 28.74 | 93.53 | 39.90 | 54.70 | 50.78 |
| COPA_KEY | Mean | 59.49 | 40.36 | 31.73 | 31.65 | 32.06 | 77.52 | 29.31 | 93.32 | 51.63 | 50.92 | 49.82 |
| | Med. | 59.39 | 42.86 | 31.10 | 32.20 | 32.50 | 83.00 | 29.22 | 93.37 | 54.81 | 51.07 | 50.00 |
| Hella._KEY | Mean | 63.10 | 49.05 | 32.93 | **34.03** | **35.48** | 81.61 | **30.15** | **94.43** | 47.79 | 51.55 | 49.88 |
| | Med. | 63.72 | 50.00 | 33.30 | **33.90** | **36.25** | 82.35 | **29.91** | **94.60** | 45.19 | 51.38 | 49.92 |
| Story._KEY | Mean | 59.49 | 40.36 | 31.73 | 31.65 | 32.06 | 77.52 | 29.31 | 93.32 | 51.63 | 50.92 | 49.82 |
| | Med. | 59.39 | 42.86 | 31.10 | 32.20 | 32.50 | 83.00 | 29.22 | 93.37 | 54.81 | 51.07 | 50.00 |
| WSC_KEY | Mean | 55.09 | 42.62 | 31.65 | 32.30 | 32.39 | 50.45 | 23.92 | 48.77 | 63.75 | 50.88 | 50.99 |
| | Med. | 54.33 | 46.43 | 31.70 | 32.50 | 32.33 | 49.50 | 23.97 | 48.69 | 63.46 | 50.75 | 50.78 |
| Wino_KEY | Mean | 62.45 | 59.76 | 31.67 | 33.76 | 34.84 | 64.57 | 26.11 | 66.65 | 48.75 | **55.79** | 51.68 |
| | Med. | 63.36 | 76.79 | 31.60 | 33.70 | 34.75 | 65.69 | 26.09 | 71.94 | 47.12 | **55.80** | 51.18 |

Table 11: Zero-shot results when training with the top-3 key tasks for each test task. The experiments are conducted on the T5-Large model. The entry at (row $i$, column $j$) denotes the model performance on test task $j$ after being trained with the top-3 key tasks of task-$i$. Mean denotes the mean performance on all prompts, and Med. denotes the median performance on all prompts. **Bold** denotes the best performance for each evaluation dataset.

| Tasks | Met. | Natural Language Inference | | | | | Sentence Completion | | | Co-Reference | | WSD |
|---|---|---|---|---|---|---|---|---|---|---|---|---|
| | | RTE | CB | ANLI1 | ANLI2 | ANLI3 | COPA | Hella. | Story. | WSC | Wino. | WiC |
| Baseline T0 | Mean | 80.72 | 67.62 | 41.09 | 37.79 | 40.38 | 91.92 | 32.03 | 97.27 | 65.96 | 57.84 | 50.14 |
| | Med. | 80.14 | 75.00 | 42.80 | 39.20 | 41.75 | 92.00 | 32.29 | 97.22 | 68.27 | 58.41 | 50.00 |
| RTE_KEY | Mean | **79.63** | 65.95 | 38.95 | 37.77 | 37.90 | 72.53 | 31.31 | 83.14 | 59.71 | 52.88 | 52.13 |
| | Med. | **79.60** | 71.43 | 38.90 | 38.00 | 37.92 | 71.96 | 31.31 | 84.02 | 59.13 | 53.04 | 52.51 |
| CB_KEY | Mean | 77.22 | 63.93 | **41.25** | **38.86** | **40.62** | 87.50 | 35.43 | 95.28 | 61.44 | 56.02 | 51.77 |
| | Med. | 76.53 | 69.64 | **41.80** | **39.80** | **42.25** | 87.00 | 36.09 | 95.24 | 62.98 | 55.56 | 52.12 |
| ANLI1_KEY | Mean | 77.00 | 60.59 | 40.73 | 37.84 | 39.13 | 87.07 | 33.45 | 96.80 | 63.37 | 54.51 | 51.05 |
| | Med. | 77.98 | 66.07 | 41.90 | 38.10 | 39.58 | 89.00 | 33.28 | 96.85 | 64.90 | 54.06 | 50.71 |
| ANLI2_KEY | Mean | 77.11 | 64.17 | 38.67 | 37.63 | 38.28 | 71.95 | 29.35 | 90.06 | 63.37 | 55.82 | 51.97 |
| | Med. | 77.26 | 67.86 | 39.10 | 37.90 | 37.92 | 71.38 | 29.23 | 92.41 | 62.50 | 56.43 | 52.19 |
| ANLI3_KEY | Mean | 58.84 | 63.93 | 37.02 | 36.19 | 37.72 | 67.14 | 25.68 | 76.24 | 42.79 | 58.63 | 51.77 |
| | Med. | 58.30 | 71.43 | 37.30 | 36.30 | 38.00 | 68.65 | 24.82 | 78.41 | 40.87 | 58.33 | 50.86 |
| COPA_KEY | Mean | 60.97 | 58.21 | 37.07 | 35.16 | 37.07 | 83.95 | 32.45 | 95.47 | 49.04 | 54.10 | 51.54 |
| | Med. | 63.36 | 60.71 | 38.30 | 35.10 | 38.58 | 88.00 | 32.71 | 95.40 | 49.52 | 53.59 | 51.33 |
| Hella_KEY | Mean | 69.42 | 65.36 | 38.85 | 37.20 | 38.50 | **87.07** | 33.66 | **97.24** | 50.38 | 55.44 | 51.03 |
| | Med. | 69.86 | 75.00 | 38.60 | 36.76 | 39.08 | **91.00** | 33.28 | **97.27** | 52.88 | 54.85 | 50.31 |
| Story_KEY | Mean | 69.42 | 65.36 | 38.85 | 37.20 | 38.50 | **87.07** | 33.66 | **97.24** | 50.38 | 55.44 | 51.03 |
| | Med. | 69.86 | 75.00 | 38.60 | 36.76 | 39.08 | **91.00** | 33.28 | **97.27** | 52.88 | 54.85 | 50.31 |
| WSC_KEY | Mean | 60.36 | 52.38 | 33.73 | 33.97 | 33.31 | 53.50 | 26.39 | 51.43 | **63.75** | 51.70 | 52.02 |
| | Med. | 59.93 | 51.79 | 33.50 | 34.00 | 33.33 | 54.00 | 26.28 | 51.36 | **63.46** | 51.62 | 51.96 |
| Wino_KEY | Mean | 66.21 | **69.88** | 34.95 | 35.17 | 27.73 | 75.27 | 25.37 | 67.48 | 50.58 | **61.50** | 50.20 |
| | Med. | 66.61 | **75.00** | 35.30 | 35.50 | 38.75 | 76.50 | 25.17 | 69.21 | 50.48 | **61.88** | 50.00 |
| WiC_KEY | Mean | 56.03 | 52.86 | 34.64 | 34.07 | 33.98 | 87.20 | 31.23 | 95.28 | 47.31 | 53.91 | 50.83 |
| | Med. | 56.68 | 57.14 | 34.70 | 34.10 | 33.75 | 91.50 | 31.41 | 95.40 | 47.60 | 52.49 | 50.24 |

Table 12: Zero-shot results when training with the top-3 key tasks for each test task. The experiments are conducted on the T5-XL model. The entry at (row $i$, column $j$) denotes the model performance on test task $j$ after being trained with the top-3 key tasks of task-$i$. Mean denotes the mean performance on all prompts, and Med. denotes the median performance on all prompts. **Bold** denotes the best performance for each evaluation dataset.

| Task Names | Met. | Natural Language Inference | | | | | Sentence Completion | | | Co-Reference | | WSD |
|---|---|---|---|---|---|---|---|---|---|---|---|---|
| | | RTE | CB | ANLI1 | ANLI2 | ANLI3 | COPA | Hella. | Story. | WSC | Wino. | WiC |
| MRPC | Mean | 49.96 | 47.74 | 33.12 | 33.21 | 32.94 | 53.9 | 23.39 | 48.06 | 61.92 | 50.4 | 50.13 |
| | Med. | 50.36 | 50.0 | 33.3 | 33.3 | 32.83 | 54.0 | 23.22 | 47.68 | 63.46 | 50.83 | 50.16 |
| QQP | Mean | 58.66 | 45.0 | 30.01 | 31.63 | 31.19 | 57.2 | 26.42 | 45.78 | 54.52 | 50.02 | 50.27 |
| | Med. | 59.03 | 48.21 | 28.4 | 31.3 | 30.17 | 56.0 | 26.77 | 45.54 | 63.46 | 50.12 | 50.16 |
| PAWS | Mean | 69.1 | 42.74 | 31.97 | 31.92 | 33.53 | 48.78 | 24.46 | 51.42 | 63.46 | 50.09 | 51.39 |
| | Med. | 70.4 | 44.64 | 32.1 | 32.0 | 33.42 | 48.0 | 24.34 | 51.9 | 62.5 | 50.12 | 51.33 |
| Hotpot QA | Mean | 57.11 | 39.4 | 33.55 | 34.41 | 33.17 | 44.02 | 24.95 | 46.82 | 59.71 | 51.57 | 50.38 |
| | Med. | 56.32 | 46.43 | 33.5 | 34.7 | 33.08 | 45.03 | 24.76 | 46.93 | 62.02 | 51.3 | 50.47 |
| Wiki QA | Mean | 63.18 | 47.26 | 29.99 | 30.88 | 33.14 | 53.74 | 22.79 | 48.84 | 59.52 | 49.72 | 51.3 |
| | Med. | 63.18 | 50.0 | 29.2 | 30.5 | 33.33 | 54.58 | 22.58 | 49.39 | 62.02 | 49.01 | 50.31 |
| Adv./DBidaf | Mean | 48.12 | 35.12 | 32.69 | 33.15 | 33.46 | 58.62 | 27.26 | 52.37 | 54.71 | 50.13 | 50.11 |
| | Med. | 47.29 | 42.86 | 33.2 | 33.3 | 33.5 | 58.0 | 27.91 | 51.52 | 61.54 | 50.04 | 50.0 |
| Adv./DBERT | Mean | 47.94 | 35.24 | 33.27 | 33.67 | 33.09 | 55.14 | 26.73 | 52.65 | 55.0 | 50.28 | 50.13 |
| | Med. | 47.29 | 50.0 | 33.3 | 33.3 | 33.0 | 55.5 | 27.37 | 51.58 | 62.5 | 50.2 | 50.08 |
| Adv./DRoberTa | Mean | 47.87 | 34.88 | 33.29 | 33.48 | 33.19 | 56.23 | 26.52 | 53.3 | 55.19 | 50.83 | 50.06 |
| | Med. | 47.29 | 50.0 | 33.3 | 33.3 | 33.0 | 56.01 | 26.54 | 52.75 | 63.46 | 50.91 | 50.0 |
| DuoRC/Self. | Mean | 54.58 | 40.12 | 32.55 | 33.33 | 32.73 | 48.92 | 26.03 | 46.25 | 47.98 | 50.2 | 48.64 |
| | Med. | 54.33 | 50.0 | 32.6 | 33.3 | 32.92 | 49.0 | 26.38 | 46.61 | 47.6 | 50.51 | 49.69 |
| DuoRC/Para. | Mean | 53.21 | 40.95 | 32.25 | 33.13 | 32.72 | 46.89 | 25.4 | 46.44 | 46.44 | 50.31 | 49.97 |
| | Med. | 52.89 | 46.43 | 32.9 | 33.2 | 32.67 | 46.58 | 25.73 | 46.66 | 44.23 | 50.2 | 50.0 |
| ROPES | Mean | 52.49 | 31.43 | 33.67 | 34.61 | 34.94 | 58.08 | 25.84 | 46.61 | 42.02 | 53.2 | 50.19 |
| | Med. | 52.71 | 32.14 | 33.9 | 35.0 | 34.67 | 56.62 | 25.95 | 46.34 | 36.54 | 53.43 | 50.0 |
| Quoref | Mean | 51.05 | 26.79 | 33.85 | 33.91 | 33.16 | 51.77 | 25.46 | 51.1 | 58.37 | 51.25 | 51.76 |
| | Med. | 50.36 | 26.79 | 33.6 | 33.7 | 33.33 | 53.0 | 25.47 | 50.13 | 62.5 | 51.22 | 50.78 |
| Cos_E | Mean | 50.14 | 38.93 | 32.68 | 33.54 | 33.34 | 66.54 | 27.2 | 63.55 | 50.19 | 51.81 | 52.23 |
| | Med. | 50.72 | 41.07 | 33.0 | 33.3 | 33.42 | 67.0 | 27.17 | 62.27 | 51.44 | 52.41 | 52.66 |
| Cosmos QA | Mean | 52.71 | 43.33 | 33.06 | 33.41 | 33.32 | 77.72 | 31.44 | 91.18 | 49.71 | 50.56 | 49.31 |
| | Med. | 53.07 | 46.43 | 33.3 | 33.3 | 33.33 | 83.0 | 31.92 | 91.39 | 54.33 | 50.83 | 50.08 |
| DREAM | Mean | 54.08 | 41.19 | 32.35 | 33.05 | 33.14 | 72.65 | 26.44 | 82.06 | 48.08 | 50.66 | 51.13 |
| | Med. | 53.79 | 50.0 | 33.2 | 33.3 | 33.5 | 74.5 | 26.35 | 82.58 | 47.12 | 50.51 | 51.57 |
| QASC | Mean | 54.91 | 43.69 | 31.71 | 33.25 | 33.64 | 53.8 | 26.5 | 65.9 | 43.37 | 52.53 | 50.42 |
| | Med. | 53.07 | 41.07 | 32.8 | 33.3 | 33.42 | 56.0 | 26.34 | 68.52 | 37.98 | 51.93 | 50.24 |
| QuAIL | Mean | 54.55 | 52.38 | 32.65 | 32.75 | 33.49 | 72.79 | 27.52 | 87.09 | 40.67 | 51.03 | 49.87 |
| | Med. | 53.79 | 62.5 | 33.1 | 33.0 | 33.83 | 73.5 | 27.56 | 87.71 | 38.46 | 50.99 | 49.92 |
| QuaRel | Mean | 47.58 | 29.05 | 33.33 | 34.59 | 33.51 | 58.04 | 25.0 | 50.8 | 44.9 | 49.69 | 50.56 |
| | Med. | 46.75 | 35.71 | 33.4 | 34.6 | 33.42 | 58.5 | 25.13 | 51.04 | 44.23 | 49.72 | 50.55 |
| QuaRTz | Mean | 54.98 | 51.9 | 31.28 | 32.64 | 33.93 | 65.2 | 25.94 | 58.57 | 36.25 | 55.99 | 52.15 |
| | Med. | 54.87 | 55.36 | 31.1 | 32.2 | 34.08 | 66.33 | 25.9 | 57.24 | 36.06 | 56.04 | 51.72 |
| SciQ | Mean | 52.96 | 20.36 | 33.15 | 32.97 | 33.28 | 50.8 | 24.63 | 45.09 | 45.19 | 51.25 | 48.71 |
| | Med. | 52.71 | 8.93 | 33.3 | 33.3 | 33.5 | 50.0 | 24.62 | 45.06 | 37.02 | 51.3 | 48.98 |
| Social IQA | Mean | 66.17 | 52.62 | 31.33 | 33.47 | 34.39 | 70.99 | 28.25 | 80.89 | 47.02 | 51.65 | 51.32 |
| | Med. | 66.79 | 62.5 | 31.4 | 33.5 | 34.25 | 73.0 | 28.18 | 82.04 | 48.56 | 51.7 | 50.86 |
| Wiki Hop | Mean | 52.96 | 34.76 | 33.53 | 33.53 | 33.29 | 52.58 | 25.82 | 48.79 | 39.52 | 50.07 | 50.03 |
| | Med. | 52.71 | 41.07 | 33.4 | 33.4 | 33.42 | 53.0 | 25.96 | 49.01 | 36.54 | 50.04 | 50.0 |
| WiQA | Mean | 52.64 | 37.98 | 32.93 | 33.46 | 33.56 | 48.96 | 21.52 | 40.5 | 37.98 | 49.6 | 51.22 |
| | Med. | 52.71 | 41.07 | 33.3 | 33.4 | 33.58 | 49.5 | 20.61 | 39.34 | 36.54 | 49.64 | 50.94 |
| Amazon | Mean | 53.65 | 49.05 | 32.47 | 32.89 | 33.38 | 55.48 | 24.36 | 48.61 | 41.25 | 49.72 | 51.3 |
| | Med. | 53.43 | 50.0 | 32.5 | 33.2 | 33.5 | 55.0 | 24.19 | 49.12 | 37.02 | 49.49 | 50.71 |
| App Reviews | Mean | 52.31 | 31.07 | 32.95 | 33.15 | 33.27 | 55.14 | 23.98 | 53.75 | 42.6 | 49.64 | 50.08 |

| Task Names | Met. | RTE | CB | ANLI1 | ANLI2 | ANLI3 | COPA | Hella. | Story. | WSC | Wino. | WiC |
|---|---|---|---|---|---|---|---|---|---|---|---|---|
| | Med. | 52.71 | 41.07 | 33.3 | 33.3 | 33.33 | 54.5 | 23.86 | 54.25 | 36.54 | 49.64 | 50.0 |
| IMDB | Mean | 55.23 | 42.86 | 32.82 | 33.47 | 33.11 | 56.81 | 24.15 | 48.78 | 38.85 | 49.68 | 50.99 |
| | Med. | 54.51 | 41.07 | 32.8 | 33.4 | 33.33 | 56.0 | 24.18 | 49.01 | 36.54 | 49.41 | 50.47 |
| Rotten Tomatoes | Mean | 54.26 | 43.45 | 32.71 | 33.17 | 33.09 | 59.19 | 23.63 | 50.58 | 40.29 | 50.31 | 53.06 |
| | Med. | 53.79 | 48.21 | 32.8 | 33.4 | 33.0 | 60.21 | 23.21 | 52.81 | 41.35 | 50.2 | 53.13 |
| Yelp | Mean | 52.71 | 34.52 | 33.25 | 33.31 | 33.54 | 51.5 | 22.67 | 45.68 | 37.4 | 50.42 | 49.83 |
| | Med. | 52.71 | 41.07 | 33.4 | 33.4 | 33.58 | 52.5 | 22.01 | 45.7 | 36.54 | 50.43 | 49.92 |
| Common Gen | Mean | 47.4 | 37.5 | 33.38 | 33.27 | 33.31 | 53.47 | 26.32 | 49.3 | 61.92 | 49.57 | 50.03 |
| | Med. | 47.29 | 50.0 | 33.3 | 33.3 | 33.0 | 54.0 | 26.71 | 49.39 | 63.46 | 49.49 | 50.0 |
| Wiki Bio | Mean | 47.83 | 42.98 | 32.93 | 33.65 | 33.06 | 50.6 | 23.08 | 44.76 | 55.58 | 49.5 | 49.84 |
| | Med. | 47.29 | 46.43 | 33.3 | 33.3 | 33.08 | 50.5 | 22.57 | 45.0 | 62.02 | 49.57 | 50.0 |
| CNN Daily Mail | Mean | 48.27 | 40.48 | 32.85 | 33.8 | 33.09 | 46.69 | 25.77 | 51.82 | 56.63 | 49.27 | 50.02 |
| | Med. | 47.29 | 50.0 | 33.2 | 33.4 | 33.0 | 45.42 | 25.85 | 51.63 | 63.46 | 49.17 | 50.0 |
| Gigaword | Mean | 53.03 | 26.31 | 33.34 | 34.19 | 33.46 | 57.45 | 23.28 | 44.5 | 49.81 | 49.98 | 49.4 |
| | Med. | 53.61 | 25.0 | 33.6 | 34.2 | 33.58 | 57.5 | 23.0 | 44.9 | 46.15 | 50.28 | 49.61 |
| MultiNews | Mean | 51.52 | 26.9 | 32.66 | 33.03 | 33.19 | 45.62 | 24.17 | 48.03 | 52.88 | 49.17 | 49.86 |
| | Med. | 52.17 | 25.0 | 32.9 | 33.4 | 33.0 | 44.0 | 24.15 | 48.0 | 57.69 | 49.09 | 49.92 |
| SamSum | Mean | 47.69 | 27.98 | 33.63 | 32.87 | 32.89 | 51.55 | 25.31 | 46.89 | 52.69 | 50.99 | 50.39 |
| | Med. | 46.93 | 28.57 | 33.6 | 33.2 | 33.08 | 52.5 | 25.34 | 46.87 | 59.13 | 50.75 | 50.08 |
| XSum | Mean | 48.81 | 42.62 | 33.43 | 33.05 | 33.08 | 57.86 | 22.35 | 45.51 | 57.31 | 49.72 | 52.45 |
| | Med. | 48.19 | 50.0 | 33.3 | 33.2 | 33.17 | 58.0 | 22.13 | 45.7 | 60.58 | 49.57 | 52.35 |
| AG News | Mean | 53.5 | 34.05 | 32.95 | 33.66 | 33.72 | 54.01 | 23.54 | 51.35 | 38.27 | 50.01 | 51.05 |
| | Med. | 53.43 | 39.29 | 33.0 | 33.6 | 33.67 | 54.5 | 23.2 | 52.65 | 37.5 | 50.2 | 50.0 |
| DBPedia | Mean | 52.78 | 35.12 | 33.24 | 33.33 | 33.67 | 54.86 | 26.45 | 52.9 | 38.46 | 51.18 | 51.29 |
| | Med. | 53.43 | 41.07 | 33.1 | 33.4 | 33.67 | 53.5 | 26.17 | 52.86 | 36.54 | 51.14 | 51.1 |
| TREC | Mean | 51.52 | 43.57 | 33.15 | 33.06 | 32.77 | 55.17 | 25.55 | 52.53 | 55.38 | 50.02 | 50.38 |
| | Med. | 51.44 | 48.21 | 33.3 | 33.0 | 33.08 | 55.0 | 25.53 | 52.86 | 63.46 | 49.72 | 50.0 |

Table 13: Experiments on T5-Large. The entry at row $i$ and column $j$ denotes the average performance when the model is trained on task $i$ and evaluated on task $j$. Mean denotes the mean performance on all prompts, and Med. denotes the median performance on all prompts.

| Task Names | Met. | Natural Language Inference | | | | | Sentence Completion | | | Co-Reference | | WSD |
|---|---|---|---|---|---|---|---|---|---|---|---|---|
| | | RTE | CB | ANLI1 | ANLI2 | ANLI3 | COPA | Hella. | Story. | WSC | Wino. | WiC |
| MRPC | Mean | 54.44 | 36.79 | 33.43 | 32.87 | 33.38 | 55.02 | 25.83 | 49.77 | 61.92 | 51.68 | 51.35 |
| | Med. | 53.97 | 37.5 | 33.6 | 32.3 | 33.17 | 56.0 | 25.62 | 49.92 | 62.5 | 51.78 | 50.47 |
| QQP | Mean | 60.87 | 46.43 | 31.07 | 32.16 | 31.72 | 58.23 | 26.23 | 47.4 | 54.23 | 50.53 | 50.13 |
| | Med. | 61.91 | 50.0 | 30.2 | 31.7 | 31.0 | 59.5 | 26.26 | 47.62 | 63.46 | 50.36 | 50.16 |
| PAWS | Mean | 74.33 | 60.6 | 35.68 | 34.91 | 35.44 | 56.65 | 27.09 | 55.0 | 61.63 | 50.75 | 51.21 |
| | Med. | 74.37 | 66.07 | 35.0 | 35.0 | 35.08 | 56.0 | 27.49 | 55.0 | 61.54 | 51.14 | 51.18 |
| Hotpot QA | Mean | 57.22 | 21.9 | 34.55 | 34.56 | 34.48 | 58.82 | 24.45 | 58.98 | 58.37 | 52.58 | 50.53 |
| | Med. | 55.6 | 10.71 | 34.3 | 35.0 | 34.25 | 60.71 | 24.2 | 58.69 | 59.62 | 52.17 | 50.24 |
| Wiki QA | Mean | 61.05 | 52.26 | 31.49 | 32.96 | 32.69 | 52.81 | 26.74 | 58.4 | 62.69 | 51.1 | 50.02 |
| | Med. | 60.83 | 51.79 | 31.0 | 32.7 | 32.58 | 52.5 | 26.89 | 58.47 | 63.46 | 51.14 | 50.0 |
| Adv./DBidaf | Mean | 48.52 | 47.14 | 32.95 | 33.6 | 33.35 | 56.35 | 26.49 | 52.68 | 56.15 | 52.17 | 50.02 |
| | Med. | 47.29 | 51.79 | 33.2 | 33.3 | 33.17 | 56.62 | 26.82 | 52.7 | 63.46 | 52.25 | 50.0 |
| Adv./DBERT | Mean | 48.81 | 37.62 | 33.27 | 33.25 | 33.46 | 60.58 | 26.45 | 54.86 | 55.38 | 51.89 | 49.91 |
| | Med. | 47.29 | 50.0 | 33.2 | 33.3 | 33.08 | 59.81 | 26.88 | 53.34 | 63.46 | 50.83 | 50.0 |
| Adv./DRoBERTa | Mean | 47.91 | 39.64 | 33.26 | 33.49 | 33.34 | 62.01 | 26.94 | 55.5 | 55.38 | 53.32 | 50.0 |
| | Med. | 47.29 | 50.0 | 33.3 | 33.3 | 33.0 | 62.75 | 27.39 | 54.36 | 63.46 | 53.59 | 50.0 |
| DuoRC/Self. | Mean | 57.11 | 43.45 | 32.83 | 33.35 | 33.51 | 55.66 | 26.71 | 45.79 | 44.13 | 52.49 | 50.0 |

| | | | | | | | | | | | | |
|---|---|---|---|---|---|---|---|---|---|---|---|---|
| | Med. | 56.86 | 48.21 | 33.4 | 33.5 | 33.33 | 55.38 | 26.75 | 46.18 | 39.42 | 52.25 | 50.0 |
| DuoRC/Para. | Mean | 53.61 | 39.17 | 33.82 | 34.15 | 33.74 | 53.83 | 26.66 | 47.66 | 38.17 | 52.83 | 50.41 |
| | Med. | 53.25 | 41.07 | 33.5 | 33.9 | 33.5 | 54.5 | 27.01 | 46.93 | 37.02 | 52.33 | 50.24 |
| ROPES | Mean | 53.65 | 41.67 | 34.2 | 34.68 | 35.64 | 69.19 | 26.13 | 50.73 | 40.38 | 57.76 | 50.31 |
| | Med. | 53.43 | 41.07 | 33.7 | 33.8 | 35.25 | 67.65 | 26.27 | 50.88 | 37.98 | 58.25 | 50.0 |
| Quoref | Mean | 47.44 | 34.17 | 33.98 | 34.52 | 33.06 | 51.1 | 25.84 | 51.26 | 60.77 | 53.76 | 49.72 |
| | Med. | 47.29 | 41.07 | 33.8 | 33.7 | 33.08 | 52.5 | 25.71 | 51.58 | 63.46 | 53.59 | 50.0 |
| Cos_E | Mean | 52.45 | 38.21 | 33.65 | 32.91 | 33.53 | 66.72 | 28.35 | 51.36 | 38.56 | 53.69 | 50.89 |
| | Med. | 51.81 | 39.29 | 33.5 | 33.0 | 33.42 | 66.83 | 29.31 | 50.94 | 36.54 | 52.88 | 51.02 |
| Cosmos QA | Mean | 55.34 | 54.64 | 35.29 | 33.2 | 33.51 | 84.57 | 33.33 | 95.06 | 47.4 | 53.92 | 51.77 |
| | Med. | 56.68 | 53.57 | 35.4 | 33.3 | 33.75 | 85.5 | 33.34 | 95.03 | 44.71 | 53.75 | 52.27 |
| DREAM | Mean | 52.38 | 37.74 | 33.53 | 32.94 | 33.66 | 76.12 | 28.57 | 80.2 | 47.4 | 52.94 | 50.2 |
| | Med. | 52.17 | 41.07 | 33.6 | 32.5 | 33.5 | 77.5 | 28.23 | 80.22 | 48.08 | 53.67 | 50.0 |
| QASC | Mean | 56.43 | 37.38 | 34.87 | 34.72 | 34.99 | 63.88 | 27.38 | 75.62 | 38.56 | 54.54 | 50.78 |
| | Med. | 56.5 | 41.07 | 34.6 | 34.7 | 34.83 | 65.5 | 27.27 | 77.12 | 36.54 | 55.09 | 50.0 |
| QuAIL | Mean | 54.12 | 49.17 | 35.12 | 34.41 | 34.32 | 79.41 | 29.28 | 89.83 | 37.12 | 54.44 | 50.13 |
| | Med. | 53.61 | 46.43 | 33.8 | 34.1 | 34.08 | 83.0 | 29.31 | 89.85 | 36.54 | 55.64 | 49.76 |
| QuaRel | Mean | 52.45 | 31.43 | 33.06 | 33.42 | 33.46 | 60.95 | 27.85 | 50.33 | 43.08 | 50.36 | 47.96 |
| | Med. | 52.71 | 33.93 | 33.1 | 33.3 | 33.5 | 58.5 | 28.49 | 50.24 | 39.9 | 50.12 | 48.04 |
| QuaRTz | Mean | 57.36 | 63.69 | 31.81 | 32.83 | 34.17 | 68.2 | 24.74 | 53.96 | 43.94 | 57.06 | 49.34 |
| | Med. | 57.4 | 69.64 | 31.4 | 32.7 | 34.0 | 65.0 | 24.48 | 52.81 | 42.79 | 56.99 | 49.76 |
| SciQ | Mean | 53.18 | 25.95 | 32.98 | 33.31 | 33.86 | 56.69 | 24.15 | 48.67 | 37.4 | 51.76 | 50.99 |
| | Med. | 52.71 | 8.93 | 33.3 | 33.3 | 33.5 | 55.38 | 23.62 | 48.53 | 36.54 | 51.46 | 50.0 |
| Social IQA | Mean | 67.69 | 51.55 | 36.76 | 35.63 | 37.85 | 72.28 | 28.83 | 80.99 | 54.33 | 54.52 | 50.91 |
| | Med. | 70.94 | 58.93 | 36.8 | 36.0 | 37.92 | 72.96 | 29.3 | 83.48 | 58.65 | 54.93 | 51.33 |
| Wiki Hop | Mean | 52.45 | 29.52 | 33.43 | 33.55 | 33.62 | 56.01 | 26.05 | 56.79 | 37.31 | 51.78 | 50.22 |
| | Med. | 52.71 | 28.57 | 33.4 | 33.5 | 33.67 | 54.08 | 26.13 | 56.97 | 36.54 | 52.25 | 50.0 |
| WiQA | Mean | 53.68 | 42.74 | 34.51 | 34.69 | 36.19 | 56.24 | 22.8 | 45.12 | 36.63 | 53.42 | 50.71 |
| | Med. | 52.71 | 39.29 | 34.5 | 34.5 | 36.0 | 56.5 | 21.86 | 45.22 | 36.54 | 53.51 | 50.0 |
| Amazon | Mean | 51.95 | 46.9 | 33.74 | 33.85 | 34.72 | 55.83 | 25.45 | 47.25 | 52.5 | 50.28 | 49.87 |
| | Med. | 52.35 | 51.79 | 33.4 | 33.9 | 34.5 | 56.0 | 25.75 | 47.25 | 59.13 | 50.12 | 50.0 |
| App Reviews | Mean | 54.37 | 31.43 | 32.94 | 33.82 | 33.09 | 58.23 | 23.74 | 56.73 | 46.92 | 49.58 | 51.88 |
| | Med. | 54.87 | 35.71 | 33.2 | 33.4 | 33.0 | 58.17 | 23.17 | 56.97 | 41.83 | 49.41 | 50.86 |
| IMDB | Mean | 55.7 | 51.9 | 34.07 | 34.11 | 34.04 | 51.85 | 26.4 | 48.66 | 48.27 | 51.25 | 48.97 |
| | Med. | 56.14 | 58.93 | 33.4 | 33.5 | 33.75 | 52.0 | 26.79 | 49.01 | 49.04 | 51.46 | 48.75 |
| Rotten Tomatoes | Mean | 54.3 | 50.95 | 33.3 | 33.39 | 33.72 | 51.62 | 26.98 | 49.93 | 39.33 | 50.66 | 51.08 |
| | Med. | 54.33 | 51.79 | 33.3 | 33.4 | 33.58 | 52.04 | 27.24 | 50.08 | 39.42 | 50.83 | 50.39 |
| Yelp | Mean | 52.71 | 32.5 | 33.38 | 33.59 | 33.38 | 53.3 | 25.65 | 46.69 | 36.54 | 49.93 | 49.97 |
| | Med. | 52.71 | 39.29 | 33.4 | 33.4 | 33.5 | 54.0 | 25.82 | 46.77 | 36.54 | 50.28 | 50.0 |
| Common Gen | Mean | 48.12 | 36.55 | 33.47 | 33.72 | 33.49 | 50.77 | 26.09 | 50.82 | 58.85 | 49.77 | 49.92 |
| | Med. | 47.29 | 48.21 | 33.3 | 33.4 | 33.17 | 51.0 | 26.24 | 50.99 | 62.5 | 49.88 | 49.84 |
| Wiki Bio | Mean | 48.38 | 34.17 | 32.73 | 33.69 | 33.08 | 55.03 | 22.72 | 46.79 | 56.83 | 51.78 | 50.2 |
| | Med. | 48.01 | 37.5 | 32.9 | 33.5 | 33.42 | 54.5 | 22.36 | 47.73 | 61.54 | 51.46 | 50.16 |
| CNN Daily Mail | Mean | 48.88 | 42.38 | 33.15 | 33.54 | 33.08 | 51.22 | 28.44 | 54.16 | 57.5 | 50.32 | 50.02 |
| | Med. | 47.65 | 48.21 | 33.3 | 33.4 | 33.0 | 51.46 | 28.94 | 54.36 | 63.46 | 50.2 | 50.0 |
| Gigaword | Mean | 53.1 | 32.38 | 33.37 | 33.84 | 34.18 | 59.66 | 23.41 | 47.15 | 37.4 | 50.67 | 49.86 |
| | Med. | 52.71 | 35.71 | 33.4 | 33.6 | 34.33 | 60.0 | 22.95 | 47.3 | 36.54 | 50.36 | 49.84 |
| MultiNews | Mean | 48.66 | 41.19 | 32.99 | 33.36 | 32.97 | 52.9 | 23.94 | 48.94 | 51.63 | 51.21 | 50.64 |
| | Med. | 48.38 | 41.07 | 33.2 | 33.3 | 32.92 | 53.04 | 24.1 | 48.85 | 57.69 | 51.07 | 50.63 |
| SamSum | Mean | 47.11 | 33.21 | 31.99 | 32.73 | 32.18 | 53.41 | 24.0 | 51.69 | 50.48 | 50.59 | 51.68 |
| | Med. | 46.57 | 35.71 | 31.7 | 32.8 | 31.83 | 52.0 | 24.01 | 52.11 | 51.92 | 50.99 | 51.25 |

| | | | | | | | | | | | |
|---|---|---|---|---|---|---|---|---|---|---|---|
| XSum | Mean | 47.04 | 43.21 | 33.31 | 34.01 | 32.94 | 52.43 | 20.31 | 48.13 | 58.65 | 49.85 | 49.98 |
| | Med. | 47.47 | 50.0 | 33.4 | 33.8 | 33.08 | 52.96 | 19.06 | 47.89 | 63.46 | 50.04 | 50.08 |
| AG News | Mean | 52.24 | 47.98 | 33.41 | 33.07 | 33.38 | 59.38 | 25.64 | 47.4 | 41.92 | 50.32 | 50.2 |
| | Med. | 52.35 | 53.57 | 33.4 | 33.3 | 33.5 | 59.31 | 25.68 | 47.35 | 43.27 | 50.51 | 50.24 |
| DBPedia | Mean | 52.74 | 18.45 | 33.3 | 33.52 | 33.39 | 61.82 | 24.87 | 47.48 | 46.25 | 50.83 | 51.14 |
| | Med. | 52.89 | 8.93 | 33.4 | 33.4 | 33.42 | 63.0 | 25.61 | 47.46 | 49.52 | 50.91 | 51.33 |
| TREC | Mean | 53.1 | 38.69 | 33.37 | 33.35 | 33.42 | 61.73 | 27.12 | 51.46 | 36.83 | 49.69 | 50.06 |
| | Med. | 52.71 | 41.07 | 33.4 | 33.4 | 33.5 | 64.5 | 27.68 | 51.31 | 36.54 | 49.57 | 50.0 |

Table 14: Experiments on T5-XL. The entry at row $i$ and column $j$ denotes the average performance when the model is trained on task $i$ and evaluated on task $j$. Mean denotes the mean performance on all prompts, and Med. denotes the median performance on all prompts.

