# OpenReview forum: "Not All Tasks Are Born Equal: Understanding Zero-Shot Generalization"
_ICLR.cc/2023/Conference — ICLR 2023 notable top 25%_

### Official Review · Reviewer_ioK2 · 2022-10-24

**Confidence:** 4
**Correctness:** 3
**Technical Novelty And Significance:** 3
**Empirical Novelty And Significance:** 3
**Recommendation:** 8

**Clarity, Quality, Novelty And Reproducibility:**

The organization and clarity of the paper could be significantly improved. Overall, I feel the reader has to work too hard to understand what was actually done and make sense of the results. I offer some specific suggestions below.

1. The study of the pairwise relationship between tasks (3.3) should be moved before section (3.2). The current order is confusing since (I assume, I would appreciate it if the authors could confirm and clarify) the pairwise relationship was what was used to select the Top-8 in Table 1.
2. Figure 1 is important, very difficult to parse, and confusing. I believe the difference between orange and red cells is the magnitude, but I'm not sure. This should be clarified. The Top-8 tasks should be highlighted. I believe the horizontal and vertical lines denote task-type groups. If so, it would help to add labels for the groups.
3. As far as I can tell, the authors use "dominating task," "dominant task," "key task," and "top-8" interchangeably. This makes the paper much more confusing than it needs to be, and the authors should settle on a single name for the concept.
4. I raise some concerns above about whether the experimental results support the paper's hypothesis as clearly as is implied, and believe some of the language in the paper should be relaxed accordingly. Systematic and well-thought-out experimentation that seeks to answer a specific hypothesis is still a valuable contribution, even if the outcome is inconclusive.

**Typos, grammar, etc**

1. In the abstract, I believe "zero-generalization" should be "zero-shot generalization"
2. In the abstract the phrase "which explains the improved zero-shot performance in recent progress" does not make sense.
3. The sentence "Zero-Shot Learning denotes no data correlated with the test set is available during the training stage." does not make sense. Perhaps the authors mean something like "Zero-Shot Learning denotes *the setting when* no data..."
4. The phrase "We stand on the shoulder of this new paradigm..." is very awkward and should be replaced. Consider replacing it with something like "We build upon previous work within this new paradigm..."
5. The authors repeatedly use the phrase "Nowadays" which is highly informal and imprecise. Recent work, recently, etc would be better a choice.
6. The citation for "Cutting down on prompts and parameters: Simple few-shot learning with language models." on page 2 is incorrect, and does not actually include the author's surname.
7. The phrase "thus hard to figure out in advance which" is not grammatically correct.
8. The sentence ending "they provide valuable commonsense and reasoning." should probably be something like "they provide valuable commonsense *knowledge* and reasoning *skills.*"
9. The tense in the sentence beginning "We will do ablation study to" should be adjusted to match the tense of the rest of the paragraph, e.g., "We do ablation studies to"
10. The sentence "Moreover, our experiments show that even though the two sets of key tasks are not exactly matched, the results demonstrate that this does not affect performance." should be revised to something like "Moreover, even though the two sets of key tasks are not exactly matched, our experiments demonstrate that this does not affect performance."

**Strength And Weaknesses:**

**Strengths**
- There is currently debate within the community about what components of multi-task prompted fine-tuning are most important for performant ZSTG. Although this work does not provide a definitive answer, the results presented here shed additional light on this issue.
- The experiments are thorough and well thought out. In general, I find the major conclusions of the paper justified and interesting.

**Weaknesses**
- Since most T0 prompts can be viewed as reformulating various NLP tasks as QA, it is not necessarily surprising that more generic QA tasks have an outsized impact on final model performance.
- Some of the results in the paper do not appear to support the paper's claims and the authors do not provide adequate discussion.
	- The results in Table 2 are confusing and don't seem to match the conclusions drawn. Only Experiment 1 shows significant performance degradation when dominant tasks are not included, and in this case, training with a random subset of tasks removed is more performant than the full set of tasks.
	- Some of the results in Table 1 should be further discussed. For example, do the authors have any explanation for why training without Top-8 improves results on ANLI2 and WiC? The bold entries in this table are very confusing. Why aren't the results for T0 Tasks w/o Top-8 on WiC in bold, this is the best-performing setting on this task, correct? I have no idea what it means for a bold result to be "comparable to or outperform the T0 baseline" because comparable is never defined.
	- The paper does not adequately justify/clarify claims related to "Specific transfer ability" and "General transfer ability" in Section 3.3. I believe the paper intends the reader to look at the diagonal blocks in Figure 1 to be convinced of the claims related to "Specific transfer ability," but I'm not sure.
- There are several issues with the proposed up/down-sampling and data-augmentation schemes.
	- The results from Table 5 do not provide consistent guidance regarding what setup works best.
	- Is there a reason the authors did not consider task-based instance reweighting as an alternative to up/down-sampling? I find this especially confusing given the authors call their technique "Task Reweighting." Task Resampling would be more clear.
	- The data augmentation scheme seems to help on average, however, the presentation seems tacked on and the technique is not sufficiently explored.
	- Why did the authors present the results as "Our Best" for the 11B model? I understand that it may have been computationally prohibitive to experiment with all settings here, however, the "Our Best" terminology implies multiple settings were explored. Including this information would be helpful for readers.
	- Overall, I'm concerned about whether these results would be statistically significant. In many cases the absolute difference between T0 (†) and T0 (\*) is larger than the difference between T0 (\*) and the best resampling + data augmentation result.
	- A number of critical hyper-parameter choices seem arbitrary. Basically, all those in the paragraph beginning "Based on our reweighting strategy" on page 8. The authors should describe how/why these specific values were selected.



**Summary Of The Paper:**

This paper studies the reason multi-task prompted fine-tuning promotes zero-shot task generalization (ZSTG) in the context of T0. In particular, the authors hypothesize that only a few tasks are crucial for ZSTG. In order to validate this hypothesis pairwise train/test performance between tasks is used to identify the top-8 "dominant" tasks in a semi-automated fashion. Fine-tuning on these tasks only, which are mostly QA tasks, is shown to outperform training on all T0 tasks. The authors additionally propose to use up/down-sampling of dominant tasks rather than the omission of non-dominant tasks, as well as a data augmentation scheme to increase the number of dominant task samples. Experiments show these changes further improve results.

**Summary Of The Review:**

Overall, I found this paper interesting and the experiments informative. However, there are some serious issues with presentation and clarity which should be addressed. Despite these issues, I still feel this paper would be a valuable contribution.

**EDIT**
The authors have updated the paper. The additional experiments in the Appendix address a number of my initial concerns and those raised by other reviewers, and I have revised my score accordingly.

---

> ### Author Response · Authors · 2022-11-19
> **Response to Reviewer ioK2**
>
> We thank the reviewer for the insightful feedback. We would like to clarify a few points as follows.
> - Response to Weakness#1: T0 prompts transform NLP tasks into "QA format". Some NLP tasks are "QA tasks" (representing certain data distribution). Both concepts are completely different--QA tasks indeed take the QA format. However, QA-formatted data are not necessarily QA tasks. Experiments (see Appendix B) show that using QA-formatted non-QA-tasks does NOT benefit zero-shot performance, proving that it is not simply the QA format that results in the zero-shot ability.
> - Response to Weakness#2:
>   - Two possible causes could be: (1) there are still QA tasks remaining in the training data. (i.e., removing the whole QA class might show something different.) (2) Some tasks show special transfer ability on the test tasks, although they do not work well on most tasks. On average, it shows that removing the tasks causes decreased performance.
>   - (1) Training without Top-8 does not improve results on ANLI2 and WiC, since both tasks are random guesses (ANLI is 33.3% and WiC is 50%) (2) Thank you for pointing out the error. We have corrected the error marking of WiC.
>   - The main difference between specific and general transferability lies in the number of tasks it works on. Tasks with general transferability can benefit a wide range of tasks, while tasks with specific transferability only benefit a small set of tasks. The tasks in the diagonal are the main type of specific transferability. Particularly, if a task only benefits a single task of another type, we also call it specific transferability.
> - Response to Weakness#3:
>   - Both upsampling and downsampling are equivalent in effectiveness, and there is no significant gap between them. However, they both achieve outperforming results than baselines. It is suggested to search for the best configuration if training computes are sufficient. Otherwise, using either of them is also accepted.
>   - The term "task-reweighting" in our paper actually means making examples of certain tasks more important or less important. Re-sampling, as far as we are concerned, is more like a technique to achieve this goal. We think there might be some understanding gap between us regarding this term, and we will probably consider revising the term in the final version after careful consideration.
>   - The main contribution of our paper is to reveal how zero-shot task generalization works and provide some related observations or insights. Data augmentation is not a focus of our paper.
>   - Good point. We only report the best result on T5-XXL, because we are unable to run all the experiments due to the limitation of computing resources. We have added detailed setting descriptions for 11B (our best) in the paper. Our best result is achieved using US+DA-T0.
>   - Both improvements brought by the process of reproducibility and our task reweighting method are statistically stable for all concerned. Also, we will release our code for the community to reproduce the results.
>   - We have appended them in Appendix A.
> - Response to Clarity, Quality, Novelty, And Reproducibility:
>   1. Thanks for your valuable suggestion, and we have rewritten this part of our paper (in Section 3.2).
>   2. We now explain the cells with different colors, and added labels for the groups now. But we do not highlight the top-8 tasks, because different people with different demands may care about different datasets.
>   3. We have unified all "dominating tasks" and "dominant tasks" into "key tasks". We do not change the representation of ``top-8'' because key tasks are not necessarily top-8 tasks. Top-8 tasks are just an approximation under our experimental conditions.
>   4. We have chosen words carefully, and expressed them more accurately.
> - Response to Typos, grammar, etc.: Thanks for pointing out the typos and misleading expressions. We have corrected all of them in our paper.

---

> > ### Comment · Reviewer_ioK2 · 2022-11-28
> > **Good updates, raising score**
> >
> > I thank the authors for their thorough reply. The additional experiments in the Appendix address a number of my initial concerns and those raised by other reviewers. Additionally, I feel the reformatting, reorganization, and rewording make the paper easier to read. I appreciate the authors' efforts here.
> >
> > In regards to data augmentation, I fully agree with the authors statement "Data augmentation is not a focus of our paper." That is what I was highlighting originally, and feel the presentation would have been stronger if it was simply omitted and left as subsequent work.
> >
> > In regards to the term "task-reweighting," while I understand the motivation, I still find the usage potentially confusing/inappropriate. Reweighting is typically used to describe reweighting terms in a loss function. For example, reweighting and resampling (up or down) are well-known approaches for dealing with class imbalance. It is also likely I'm being overly pedantic here, as this was not noted by any of the other reviewers.
> >
> > Lastly, while many of the typos have been fixed I still noticed several when reviewing the revised paper. A few I noted are below.
> >
> > 1. The sentence "This section understands how multi-task training contributes to zero-shot generalization." is probably intended to be something like "This section *seeks to understand* how multi-task training contributes to zero-shot generalization."
> > 2. There is a space before a comma on page 1 in "an improved method, task reweighting , which".
> > 3. In the appendix, the citation in "GPT-Neo-1.3B (?)" is missing.
> >
> > I would encourage the authors to take a final editing pass over the paper.

---

> > > ### Author Response · Authors · 2022-12-01
> > > **Response to Reviewer ioK2**
> > >
> > > Thank you for your constructive feedback.
> > >
> > > For the term "task-reweighting", we would like to follow the standard convention and change it into "task-resampling", to align with a common understanding. We agree that "task-resampling" is a more precise and unambiguous term.
> > >
> > > For typos, we will further revise the paper for the final version.

---

### Official Review · Reviewer_wgeL · 2022-10-24

**Confidence:** 4
**Clarity, Quality, Novelty And Reproducibility:** The paper is well written, with good …
**Correctness:** 3
**Technical Novelty And Significance:** 2
**Empirical Novelty And Significance:** 2
**Recommendation:** 6

**Strength And Weaknesses:**

Strength:
1. Good writing with clear logic.
2. The finding " training on a small number of key tasks beats using all the training tasks" looks interesting.

Weakness:
1. The finding that a small set of key tasks can beat the mixture of all tasks is not so surprising.
2. The paper states "key tasks are mostly QA tasks" and the analysis is that "some QA tasks show general transfer ability", and "some QA tasks require some simple reasoning ability in the general domain…difficult to learn this knowledge in pretraining stage". However, the experiments were only conducted on 8 task families(38 tasks), and 21 of 38 tasks were in QA task type. The conclusion may be inconsistent with considering more task families such as dialogue, semantic parsing, and commonsense tasks. The experiment is incomplete.
3. In my understanding, the dataset of key tasks will be duplicated by 5 times via the task reweight mechanism. However, how much time will the general data set be used in T0 baselines? Is the comparison between the model trained on duplication data and a regular T0 unfair? (i.e. Will the resulting improvement come from more duplicated data rather than the key knowledge it contains?)
4. The task reweight method is based on the pairwise generalization results (upsampling based on positive transfer, and downsampling based on negative transfer). However, ExT5(https://openreview.net/pdf?id=Vzh1BFUCiIX) (Table 3) shows that the best-effort mixture set based on the pairwise positive transfer did not outperform a random selection of tasks, which seems contrary to the statement in this paper. Besides, it would be better if the author compare a baseline model doing upsampling and downsampling based on random selection(keep the duplication and downsample times the same as the proposed method).
5. The result in Table2 is inconsistent with the claim. Only experiment1 shows a significant performance degradation when removing key tasks.
6. This submission is more like a technical report and the inspiration is limited.

---------------After Rebuttal ----------------

I thank the authors for their response. The authors have addressed most of my concerns and added new experiments that have improved the paper. I have raised my score.



**Summary Of The Paper:**

This paper conducts experiments to understand how multi-task learning for zero-shot generation works. The conclusion is that some key tasks (QA) dominate the generalization. The paper further proposes a simple task reweight method to upsample important tasks and downsample redundant tasks based on pair-wise transfer ability across tasks.


**Summary Of The Review:**

This paper has certain contributions, but I do not think the paper is enough for getting accepted by ICLR.

---

> ### Author Response · Authors · 2022-11-19
> **Response to Reviewer wgeL**
>
> We thank the reviewer for the insightful feedback. We would like to clarify a few points as follows.
> 1. Response to Weakness#1: To the best of our knowledge, this finding has never been rigorously demonstrated before. Some work [1] thought that the prompted multi-task training benefits from "responding to instructions", which highlights a unified format of training data. Combined with the fact that "a single task cannot help the model learn responding to instructions well", training with only one/several dataset(s) should not achieve good performance. However, our experiments show that key tasks are extremely important, which highlights the importance of knowledge or features in the training data. Our conclusion would urge the community to rethink the essence of multi-task zero-shot learning.
> 2. Response to Weakness#2: Good point. We additionally experimented with a totally different mixture of datasets---there are 43 training tasks in total and QA tasks only account for 7/43 among them. Results prove that our conclusion still holds. We have reported this new experiment in Appendix C.3.
> 3. Response to Weakness#3: That is a valuable point. Note that though we have performed upsampling/downsampling, the total number of data is much less than those used by T0, proving that it is NOT duplicated data that results in the improvements. Please refer to Appendix B.1 for detailed statistics. What's more, it is fair since we do not introduce new data into the experiment. Theoretically, duplicate data may only require training fewer epochs but will not change the performance, as long as they all converge. Give a simple example, let the original dataset A, and duplicated dataset B where all data in A are duplicated by five times. Then, training on dataset B for one epoch will be the same as training five epochs on dataset A.
> 4. Response to Weakness#4: ExT5 and our method are entirely different and cannot be compared, whereas ExT5 explores finetuning performance while we focus on zero-shot ability. Intuitively, for ExT5, since train data of the same task are used during finetuning, some knowledge is learned by the model. At this time, tasks that provide similar knowledge are not necessary anymore, and complementary tasks will be more important. However, the zero-shot test task requires positive-transfer tasks to provide the knowledge it requires.
> 5. Response to Weakness#5: Two possible causes could be: (1) there are still QA tasks remaining in the training data.(i.e., removing the whole QA class might show something different.) (2) Some tasks show special transfer ability on the test tasks, although they do not work well on most tasks. On average, it shows that removing the tasks causes decreased performance.
> 6. Response to Weakness#6: Same as Response to Weakness#1
>
> [1] Wei J, Bosma M, Zhao V Y, et al. Finetuned language models are zero-shot learners[J]. arXiv preprint arXiv:2109.01652, 2021.

---

### Official Review · Reviewer_nkcG · 2022-10-25

**Confidence:** 3
**Correctness:** 3
**Technical Novelty And Significance:** 2
**Empirical Novelty And Significance:** 3
**Recommendation:** 5

**Clarity, Quality, Novelty And Reproducibility:**

The experimental sections have been clearly written at least to point out the main message. There is not much novelty in the experimental setup, other than the careful grid search over the multiple experimental settings.

**Strength And Weaknesses:**

The major strength of the paper is the extensive experimentation to show that few tasks are dominant in the multi-task training of the T5 model. Furthermore, the authors come up with a novel strategy to find the optimal mix of different tasks during multi-task training.

One of the questions is whether the result depends on the usage of T5-large. Will the results change if we change the size of the T5 model, or move to encoder-only or decoder-only models?

Furthermore, I have the following concerns/questions:

(a) How were the Top-8 tasks decided in table 1? Did the authors try with all possible subsets of 8 tasks and find the Top-8 tasks to be the best performing? Also, how did the authors decide to use "8" tasks and not fewer?

(b) For the performance scores in table 1 and figure 1, are the same hyperparameters used in all the experiments? If so, can the authors comment on the possibility of the findings being dependent on the hyperparameters used for multi-task training?

(c) What does the ID column represent in table 2? Moreover, how does the performance change when $D_{rand}$ belongs to a specific task type (e.g. {MRPC, QQP, PAWS}, { Adv./DBidaf, Adv./DBERT, .. } )

Furthermore, how does the performance change when $D_key$ represents a subset of the extractive QA and multi-choice QA tasks?

(d)  In section 4.1, when identifying the key tasks, the authors observe the cross-generalization performance of each task independently, i.e. they measure g(A) for each A independently. However, in section 3.2, they measured the performance of the Top-8 tasks as a group (same in section 3.4). Hence, do the authors believe in the modularity of the function g, i.e. whether g({A, B}) = g(A) + g(B), where A and B are two different tasks? If g({A, B}) >> g(A) + g(B), then one can't identify that tasks A and B are both important for multi-task training by looking at g(A) and g(B) independently.

(e) In the experimental setup, how were $TH_1$, $TH_2$, $N_d$, and $N_u$ decided?


**Summary Of The Paper:**

The paper shows on T5-large that in multi-task training, the model can perform better zero-shot generalization from training on a few QA tasks compared to training on all the tasks. Furthermore, the authors observe a drop in performance, if the key tasks are removed from the multi-task training dataset. They show that the key tasks show better transferability to other tasks. They hypothesize that the key tasks contain information not available to the model during pretraining. Finally, they come up with a strategy to identify the optimal mix of the tasks to be used during multi-task training.

**Summary Of The Review:**

Overall, my scores are on the borderline. I am unsure of how much the results can hold true for other models.

---

> ### Author Response · Authors · 2022-11-19
> **Response to Reviewer nkcG**
>
> We thank the reviewer for the insightful feedback. We would like to clarify a few points as follows.
> 1. The results will NOT change if we move to base models of different scale and architecture. We have added results on T5-3B and GPT-Neo in Appendix C.
> 2. Response to (a): To clarify how the top-8 tasks are decided---(1) We first get the pair-wise task performance (as Figure 1 shows), i.e., training on one task and evaluating how it performs on the other task. In this way, we get the transferability between any pair of tasks. (2) For each test task, we select the top-3 train tasks that perform best, obtaining several triple sets. (3) Key tasks are those that appear at least twice in the triple sets, which exactly results in 8.
> 3. Response to (b): Hyper-parameters are decided according to practical considerations. Details are in Appendix A. It takes much fewer steps to converge for small datasets; thus, we train fewer steps on those small datasets for the sake of time. We use the same hyper-parameters for datasets of similar size. We did preliminary experiments in terms of different hyper-parameters, and the results show that it does not cause much difference (less than 1 point) when the training step is large enough. We believe our findings are not sensitive to hyper-parameters.
> 4. Response to (c): As is pointed out in our paper, $D_{key}$ represents a subset of the extractive QA and multiple-choice QA tasks. We do agree that it would be interesting if we experiment with $D_{rank}$ belonging to a specific task type. We will add these results in our future version.
> 5. Response to (d): It is true that our method would miss cases where g({A, B})>>g(A)+g(B). (which we leave for future work). Our experiments proved that only identifying part of key tasks is enough to achieve significant improvements. This strategy is simple and effective. Besides, we practically performed smooth upsampling/downsampling instead of simply removing all unimportant tasks, so the influences of those missing tasks are not completely eliminated.
> 6. Response to (e): We have added details of how we determine hyper-parameters in Appendix A. In short, we predefined them according to some preliminary experiments based on T5-Large, and then directly applied them to larger-scale models.

---

### Official Review · Reviewer_DTcG · 2022-10-25

**Confidence:** 4
**Correctness:** 4
**Technical Novelty And Significance:** 4
**Empirical Novelty And Significance:** 3
**Recommendation:** 8

**Clarity, Quality, Novelty And Reproducibility:**

I found this paper easy to read, and very novel. It seems like the takeaways would be straightforward to reproduce.

**Strength And Weaknesses:**

Strengths:
- Overall, this was a really interesting paper, both from empirical (higher metrics) and exploratory (which QA tasks are better? why?) points of view. We often treat benchmark tasks as opaque and indistinguishable, so this paper was a refreshing look at the actual content and effect of specific tasks, as well as their interdependence and similarity.
- The empirical results are compelling, and have solid practical implications for future model development (e.g., reweighting, augmentation, etc)
- The side-by-side analysis of QA task datapoints (Table 3) was really useful. It would be great to have a couple of examples for each of the rest of the datasets.

Weaknesses (and some suggestions/questions):
- I wish there was more exploration of the task dataset similarities and differences, and how that might influence the results. This could be something as simple as average distance between embeddings of dataset A and B (though there are probably better metrics. Even text length distributions would be interesting). Does similarities and differences account for any of the task transferability?
- Similarly, in Table 3, the authors say that the biggest difference between SocialIQA/CosmosQA and Wiki_hop/WiQA are their knowledge domain, but it seems like the text format would have a significant impact as well. That is, Wiki_hop/WiQA seem much more artificially-constructed than SocialIQA/CosmosQA, which are more in natural language that might be more broadly applicable.
- What were the original sizes of each task dataset (I might have missed this in the paper), and are you making sure each task has the same number of datapoints (before the explicit reweighting experiment)? Also, other per-dataset metrics would be useful (even just something like lexical diversity)


**Summary Of The Paper:**

The authors explore which types of tasks are most helpful for pretraining a model for universal zero-shot task performance. They find that QA tasks are most helpful. They then go on to see how generalizable this is (ie, are QA tasks just helpful for other QA tasks? Answer: no!) Finally, they propose a method to re-sample training tasks for better downstream performance.

**Summary Of The Review:**

I'd recommend accepting this paper-- it presents a method for solid empirical improvement on few-shot learning, and fleshes this out with a deep qualitative and quantitative analysis on the influence of individual task datasets.

---

> ### Author Response · Authors · 2022-11-19
> **Response to Reviewer DTcG**
>
> We thank the reviewer for the insightful feedback. We would like to address several points as follows.
> 1. That is a great aspect of exploration! (1) We have experimented with using semantic embeddings to measure the distance between tasks. However, we found that using the distance between the average sentence embeddings of two datasets does not give good guidance for the transfer ability. (2) We have also explored a few points in terms of task similarities and differences (e.g., sentence length, task domain, task family, etc.). The facts are that the impacts of these task features are highly coupled, which cannot be analyzed independently. Specifically, the correlation between every single factor and generalization performance is low.
> 2. The ``text format''' is indeed an important aspect. The analytical statement is not rigorous. We humbly accept the suggestion. However, to emphasize, it does not affect the effectiveness of our proposed method, since we do NOT rely on any individual task feature to determine key tasks, but use a pre-detection algorithm according to overall performance.
> 3. Good question. We add statistics (including the suggested lexical diversity) of original datasets in Appendix B.1. The original tasks are different in size, which is the same as T0. It is not necessary to start from a setting where all tasks are the same in size. We found a better composition ratio of multiple tasks than the current best-performed setting (i.e., T0.)

---

### Decision · Program_Chairs · 2023-01-20

**Decision:**

Accept: notable-top-25%

**Justification For Why Not Higher Score:**

This paper is useful and interesting, but will probably not be transformative. That is, it has insights that are worth sharing and advertising, but I bet the heuristic reweighting scheme won't be adopted because it's overly complicated and something else that works better will probably come along. However I think the core insights are worth sharing.

**Justification For Why Not Lower Score:**

There was consensus among reviewers for acceptance. I think the paper's findings around which tasks are most useful for zero-shot generalization should be shared with a wider audience.

**Metareview: Summary, Strengths And Weaknesses:**

This paper studies zero-shot task generalization of multitask prompted models (such as T0). The paper first identifies that a small subset (the QA datasets) of T0's training datasets are sufficient to attain strong zero-shot generalization capabilities. Motivated by this finding, a heuristic reweighting scheme is proposed where a training dataset is upweighted if it results in positive transfer to another training dataset of a different task type. Using this reweighting scheme improves zero-shot generalization. Reviewers generally agreed that this paper had interesting results and was worth publishing. While there were some questions on methodology and requests for clarification, these issues were settled after the rebuttal.

**Note From Pc:**

if the above contains the word "oral" or "spotlight" please see: "oral" presentation means -> notable-top-5% and "spotlight" means -> notable-top-25%. As stated in our emails, we are disassociating presentation type from AC recommendations